

**Transboundary ozone pollution across East Asia: daily**
**evolution and photochemical production analysed**
**by IASI+GOME2 multispectral satellite observations and**
**models**
**Juan Cuesta[1], Yugo Kanaya[2], Masayuki Takigawa[2], Gaëlle Dufour[1], Maxim**
**Eremenko[1], Gilles Foret[1], Kazuyuki Miyazaki[2] and Matthias Beekmann[1]**
[1] {Laboratoire Inter-universitaire des Systèmes Atmosphériques (LISA), UMR7583,
Universités Paris-Est Créteil et Paris Diderot, CNRS, Créteil, France}
[2] {Japan Agency for Marine-Earth Science and Technology, Yokohama, Japan}
Correspondence to: Juan Cuesta (cuesta@lisa.u-pec.fr)
**Abstract**
We characterize a transboundary ozone pollution outbreak transported across East Asia in
early May 2009 using new multispectral satellite observations of lowermost tropospheric
ozone in synergy with other satellite data and models. Our analysis is focused on the daily
evolution of ozone pollution plumes initially formed over the North China Plain (NCP) and
their transport pathways over Northern China, Korea, Japan and the surrounding seas. A main
aspect of the study is an estimation of the contribution of photochemical production of ozone
along transport using the ratio of ozone to carbon monoxide enhancements with respect to
background levels derived from satellite data and also from chemistry-transport models.
A key contribution of the analysis is the use of new satellite data offering unprecedented skills
to observe the horizontal distribution of lowermost tropospheric ozone over East Asia on
daily basis, with a multispectral approach called IASI+GOME2. These satellite observations
are in good agreement with ozonesondes, with low mean biases (3%), a precision of about
16%, a correlation coefficient of 0.85 and practically the same standard deviation for a
comparison based on 2 years of data from 46 launching stations distributed worldwide, during
all seasons. A similar agreement is also found over East Asia. Moreover, IASI+GOME2





offers a unique capacity for observing the evolution of near surface ozone during pollution
outbreaks (with 5 % bias and 0.69 correlation), according to a comparison with surface in situ
measurements during 2 major ozone events over several Japanese Islands. Single-band ozone
retrievals, as those from IASI in the thermal infrared, do not capture such variability.
Using IASI+GOME2, we put in evidence that i) ozone pollution plumes are transported by an
anticyclonic circulation around the Yellow Sea from the NCP to Northern China, Korea and
Japan, co-located with carbon monoxide plumes, ii) over Northern China the plume splits into
two pollution filaments with one mixing with freshly emitted pollutants and iii) ozone is
produced every day of the event accounting for an enhancement in concentration during
transport across East Asia of up to ~84 % with respect to that produced over NCP. This
estimation is done according to monotonically increasing values during 7 days of the ratio of
ozone to carbon monoxide enhancements within the transported pollution plumes from about
~0.25 over the NCP to ~0.46 over the Pacific south of Japan.

## 1 Introduction

Air pollution is now the world's largest single environmental health risk, causing 7 million
premature deaths worldwide every year (Lelieveld et al., 2015; World Health Organisation -
WHO, 2016). About 4.3 million of these deaths are related to ambient air pollution, from
which 2.6 million deaths per year occur over Southeast and East Asia as result of exposure to
the world's largest air pollution-related burden (WHO, 2016). East Asia, and in particular
China, experienced rapid economic growth (up to a factor 40 of the gross domestic product
since the 80's) and extensive urbanization during the last decades. Accordingly,
anthropogenic pollutant emissions have largely increased, making China one of the largest
pollution source regions in the world (Lu et al., 2011, Wang et al., 2013). In the main Chinese
megacities, ambient concentrations of the most harmful pollutants, such as tropospheric ozone
($O_3$) and particulate matter (PM), largely exceed the thresholds recommended by WHO (Chai
et al., 2014). Air pollution originating from East Asia is also a worldwide-shared concern. It
can be transported and undergo chemical transformations far beyond country boundaries
within a day and around the hemisphere within one or two weeks, having a significant impact
on the budget of tropospheric pollutants at the intercontinental scale (e.g. Lin et al., 2010). For
example, current trends of ozone concentrations at the surface over Japan show a significant



increase, despite strong local controls of pollution emissions in the last decades, and probably
related to transboundary transport (Akimoto et al., 2015). Similarly, transcontinental transport
of Asian pollution probably explains the absence of reduction in ozone background levels
over the United States or Europe, despite local efforts for reducing the emissions of its
precursors (e.g. Dentener et al., 2010, Verstraeten et al., 2015).
The dramatic damages caused by East Asian air pollution at regional and intercontinental
scales strongly request for thorough monitoring of pollutant emissions both near the sources
and downwind from these regions, where secondary pollutants are photo-chemically
produced. However, the record of surface network observations of air pollution over China is
very limited, being openly available only since 2013 (Wang et al., 2014). On the other hand,
forecasting East Asian air pollution with chemistry-transport models is hampered by two
factors: insufficient surface observations for validating the simulations (particularly over the
East China Sea) and the lack of precision of the emission inventories, which are unable to
reflect the rapid changes in Chinese economy and the complexity of their emissions (Wang et
al., 2015).
Satellite observations offer a great potential for filling the observational gap of air pollution
over East Asia and overcome the limited spatial coverage of ground-based measurements.
Nevertheless, measuring ozone pollution from space is a challenging issue. Standard single-
band ozone retrievals cannot provide quantitative information at the planetary boundary layer
(PBL), but at the free troposphere located above. Spaceborne spectrometers operating in the
UV, like OMI (Ozone Monitoring Instrument, Levelt et al., 2006) and GOME-2 (Global
Ozone Monitoring Experiment-2, EUMETSAT, 2006), have been used to derive tropospheric
ozone observations with sensitivity around 5-6 km of altitude (e.g. Liu et al., 2010, Cai et al.,
2012). Thermal infrared (IR) space-borne instruments, like IASI (Infrared Atmospheric
Sounding Interferometer, Clerbaux et al., 2009) on-board the MetOp satellites, have shown
good performance for observing ozone in the lower troposphere, but with sensitivity peaking
at 3 km of altitude at lowest (e.g. Eremenko el al., 2008; Dufour et al., 2012). Recently, a new
multispectral approach called IASI+GOME2, combining IASI observations in the IR and
GOME-2 measurements in the UV, allowed the first spaceborne observation of the full
horizontal structure and concentration of ozone plumes located near 2 km of altitude, for a
moderate European pollution outbreak (Cuesta et al., 2013). This approach offers the unique
capacity to observe the horizontal distribution of ozone in the lowermost troposphere (LMT),





hereafter defined as the atmospheric layer between the surface and 3 km of altitude above sea
level (asl). Similarly, the multispectral combination of TES (Tropospheric Emission
Spectrometer, Worden et al, 2007) and OMI measurements, respectively in the IR and UV,
has also shown an enhancement of sensitivity below 700 hPa (Fu et al., 2013), but with very
limited horizontal coverage (pixels longitudinally spaced by about 2000 km on the same day).
Simultaneously monitoring several air pollutants may offer useful insights on the origin and
evolution of ozone pollution. For example, high concentrations of both ozone and carbon
monoxide (CO) suggest an anthropogenic origin of the air masses, as CO is a primary product
of traffic and industrial emissions and is formed by oxidation of anthropogenic hydrocarbons.
Its lifetime is about 2 months (e.g. Logan et al., 1981). Tropospheric ozone-enriched
airmasses with background concentrations of CO and low water vapour levels are probably
related to downward transport from the stratosphere and the Upper-Troposphere/Lower-
Stratosphere (UTLS) region. The ratio between the enhancements of $O_3$ and CO with respect
to the background levels allows examining the production of ozone from combustion by-
products (nitrogen oxides - $NO_x$, hydrocarbons and CO) by photochemical processing of air
parcels during a few days to a week (e.g. Parrish et al., 1993, Chin et al., 1994, Mauzerall et
al., 2000). This approach may however underestimate ozone production along transport since
CO may not only be directly emitted but also produced by oxidation of hydrocarbons (Chin et
al., 1994; Gao et al., 2005). This ratio has been mainly estimated using in situ measurements
at several ground-based sites (Chin et al., 1994), from aircrafts (Price et al., 2004), model
simulations (Maurezall et al., 2000) and in a few cases with satellite data mainly sensitive at
the free troposphere (Zhang et al., 2006; Kim et al., 2013; Dufour et al., 2015). In addition,
high abundances of ozone precursors, such as nitrogen dioxide ($NO_2$) and volatile organic
compounds as formaldehyde ($CH_2O$), may be linked to higher photochemical production of
$O_3$, depending on the regime of ozone atmospheric production (i.e. either limited by the
availability of nitrogen oxides $NO_x$ or volatile organic compounds VOC).
In the present paper, we characterize the daily evolution of a major ozone outbreak across
East Asia in early May 2009, using the new multispectral satellite approach IASI+GOME2 in
synergism with chemistry-transport and meteorological models as well as other observations
(CO, $NO_2$, $CH_2O$, etc.). We present the first observational description of the transport
pathways of ozone plumes from satellite measurements at the LMT (below 3 km of altitude)
over East Asia and we analyse the processes controlling the lowermost tropospheric ozone



burden during this event (i.e. photochemical production and downward transport from the
stratosphere). Our study uses the ratio between the enhancements of $O_3$ and CO to
characterize the Lagrangian production of ozone along transport across East Asia, derived for
the first time from ozone satellite data sensitive at the LMT. First, the paper presents the
datasets used in the study and a quality assessment of the IASI+GOME2 ozone observations
by comparing them with in situ measurements performed by ozonesondes and also by surface
stations (section 2). This comparison illustrates the unprecedented capacity of IASI+GOME2
to observe from space the variability of surface ozone concentrations. Section 3 describes the
regional distribution of ozone plumes and the meteorological conditions during each day of
the pollution outbreak. Then, we focus on the Lagrangian evolution of one of the major ozone
plumes, analysing the possible enhancement of ozone concentrations by photochemical
production during transport (section 4). A summary is provided in section 5.
**2   Datasets description**
**2.1   Satellite observations of lowermost tropospheric ozone: IASI+GOME2**
The multispectral satellite approach IASI+GOME2 is designed for observing lowermost
tropospheric ozone by synergism of thermal IR atmospheric radiances observed by IASI and
UV earth reflectances measured by GOME-2. Both instruments are onboard the MetOp
satellite series (in orbit since 2006 and expected until 2022) and they both offer global
coverage every day (for MetOp-A around 09:30 local time, LT) with a relatively fine ground
resolution (12 km-diameter pixels spaced by 25 km for IASI at nadir and ground pixels of 80
km × 40 km for GOME-2). As described in detail by Cuesta et al., (2013), IASI+GOME2
jointly fits co-located IR and UV spectra for retrieving a single vertical profile of ozone for
each pixel. The horizontal resolution corresponds to that of IASI, using for each pixel the UV
measurements from the closest GOME-2 pixel (without averaging). Spectra and Jacobians in
the IR and UV are respectively simulated by the KOPRA (Karlsruhe Optimized and Precise
Radiative transfer Algorithm; Stiller et al., 2002) and VLIDORT (Vector Linearized Discrete
Ordinate Radiative Transfer; Spurr, 2006) radiative transfer codes. The effects of clouds and
aerosols are partially taken into account by iteratively adjusting offsets for each of the 7
spectral micro-windows (between 980 and 1070 cm$^{-1}$) used in the IR and effective surface
albedos and cloud fractions in the UV (2 micro-windows between 290 and 345 nm). Only
measurements with cloud fractions below 30% are used (as determined by the FRESCO



algorithm, Koelemeijer et al., 2001). Ozone profiles are retrieved by a constrained least
squares fitting method using a Tikhonov-Phillips-type regularisation (Tikhonov, 1963).
Constraint strengths vary with altitude and are optimised for enhancing sensitivity to
lowermost tropospheric ozone while keeping acceptable total retrieval errors (in the order of
20 % for the LMT).
Here, we use an updated version of the IASI+GOME2 product, with only minor changes with
respect to that of Cuesta et al. (2013). Ozone profiles are retrieved at the vertical grid between
the surface and 60 km of altitude asl (above sea level), with steps of 1 km, 2 km and 5 km,
respectively below 26 km asl, between 26 and 30 km asl and above. Three a priori ozone
profiles derived from the climatology of McPeters et al., (2007) are used, corresponding to the
average over 20-30°N, 30-60°N and 60-90°N, representative of tropical, mid-latitude and
polar conditions. These three a priori profiles are used for IASI pixels with tropopause heights
(determined by the temperature vertical profile) above 14 km, between 14 and 9 km and
below 9 km, respectively.
IASI+GOME2 products include vertical profiles of ozone, partial columns, averaging kernels
(representing sensitivity of the retrieval to the true atmospheric state), error estimations and
quality flags. From 2017, global scale IASI+GOME2 retrievals are routinely produced and
publicly available by the French data centre AERIS (http://www.aeris-data.fr and http://cds-
espri.ipsl.fr).

### 20   2.1.1  Validation of IASI+GOME2 at the LMT against ozonesondes

An assessment of the quality of IASI+GOME2 for retrieving LMT ozone is presented in
Figure 1. It is based on a comparison of IASI+GOME2 retrievals and ozonesondes
measurements, for the first time spread at the global scale and for all seasons during two
years. We consider ozonesondes launched from 46 different sites (spread worldwide from
69°S to 83°N and 171°W to 152°E) along the years 2009 and 2010 (provided by the World
Ozone and Ultraviolet radiation Data Centre - WOUDC, http://www.woudc.org). Vertical
resolution of the ozonesonde profiles is about ~150 m and their errors are about ± 5%
(Deshler et al., 2008). Coincidence criteria are spatial co-localization within ± 1-degree
latitude/longitude (as for Keim et al., 2009; Dufour et al., 2012; Cuesta et al., 2013) and a
time frame of 12 h from the MetOp-A morning overpass (at 09:30 LT). These differences in
time and location induce part of the random differences between the satellite retrievals and the





ozonesondes. The comparison is made for each ozonesonde with the average of collocated
satellite retrievals (thus partly reducing random errors). To account for the retrieval
sensitivity, we calculate "smoothed" ozonesonde measurements (indicated in Fig. 1 as
"SONDE*AVK") by interpolating at the satellite retrieval vertical grid (with 1 km-vertical
resolution below 26 km), convoluting with each of the averaging kernels (AVKs) of the
collocated satellite retrievals and then taking the average. Only quality-assured retrievals of
IASI+GOME2 are used (discarding too high fitting residuals, cloud fraction above 30 %,
aberrant retrievals of surface temperatures, ozone profiles or AVKs). After cloud screening
and quality-checks, the number of sondes with coincident IASI+GOME2 data used for this
comparison is 1035.
The comparison at the worldwide scale shows a good agreement of IASI+GOME2 and
ozonesondes in the lowermost troposphere, with a weak mean bias (-3 %), a good correlation
(0.85), a very similar variability (a ratio of standard deviations of ~1.0) and a precision of 16
% (estimated as the root-mean-squared difference between the two datasets, see Fig. 1a).
These good results are very similar to those obtained in a first validation exercise over Europe
during the summer of 2009, with practically the same correlation, precision, variability and
weak bias (Cuesta et al., 2013). As this paper focuses on East Asia, we also present the
comparison for all sondes available over this region in 2009-2010 (112 sondes after cloud
screening and quality checks), launched from the 3 Japanese sites of Sapporo, Tateno
(Tsukuba, near Tokyo) and Naha (Fig. 1b). In this case, IASI+GOME2 shows similarly good
performance, with a weak bias (3 %), the same variability as that of sondes, a precision of 13
% and a good correlation (0.76) slightly lower with respect to the global comparison
(probably partly linked to a lower variability in the measurements).

### 2.1.2  Capacity of IASI+GOME2 to observe near-surface ozone

An additional quality assessment is shown in Figures 2 and 3, which evaluates the capacity of
IASI+GOME2 to observe near-surface ozone pollution over East Asia. IASI+GOME2
retrievals at the LMT are compared with in situ measurements at the surface, from 10 stations
of the EANET/GAW (Acid Deposition Monitoring Network in East Asia / Global
Atmosphere Watch, http://www.eanet.asia) networks over East Asia and one station at Fukue
Island (32.8°N, 128.7°E e.g. Kanaya et al., 2016) operated by JAMSTEC institute (see the
location of all these stations in Figs. 3a-b). These stations are representative of background
rural environment over several Japanese islands. We consider the 2 major ozone pollution





events observed at the surface over Japan during the springtime 2009, respectively on 4-9
April and 4-9 May 2009 (as suggested by higher ozone surface concentrations measured by
EANET/GAW/JAMSTEC). Co-localisation in time and space is assumed within ± 1 h and ±
1° degree latitude/longitude, respectively. The comparison is made between the surface in situ
hourly measurements for the satellite overpass time and the average of collocated satellite
retrievals. Table 2 presents the results of the comparison of all coincident satellite retrievals
(both for IASI+GOME2 and IASI only) and surface measurements (the 3 datasets are
available for each coincidence). We consider 2 sets of surface measurements in order to
account for IASI+GOME2 LMT sensitivity (which peaks near 2 km asl over land): i) those
corresponding to vertical gradients $\Delta O_3^{surf.-2km}$ between the surface and 2 km of altitude lower
than ± 20 ppb and ii) the whole dataset (respectively 44 and 52 coincidences). The gradient
$\Delta O_3^{surf.-2km}$ is estimated from analyses of the tropospheric ozone distribution derived from the
CHASER chemistry-transport model (see section 2.3). Figures 2 and 3 shows respectively the
scatter of points for the case with limited $\Delta O_3^{surf.-2km}$ (similar to that for all measurements) and
an illustration of the horizontal distribution of ozone satellite retrievals and surface
observations.
Figures 2a and 3a show a good agreement between IASI+GOME2 and the ozone in situ
observations at the surface. To the authors' knowledge, this is the first time that such
agreement is found for a satellite retrieval of ozone and surface measurements. IASI+GOME2
observations show a fairly good correlation (up to 0.69), a mean bias of -5 %, a precision of
20 % (similar to the retrieval error of IASI+GOME2 at the LMT) and a similar standard
deviation with respect to the surface in situ measurements. Slightly lower correlation (0.63) is
remarked when comparing all observations (regardless $\Delta O_3^{surf.-2km}$) of the period (52 cases),
but the agreement remains fairly good. This is illustrated for one of the days in the Figure 3a,
where IASI+GOME2 clearly captures the high concentrations of the ozone plumes over the
Japan Sea, south Japan and the Pacific. The multispectral satellite approach is also capable to
observe some of the horizontal gradients within the plume, such as the relatively lower ozone
concentrations over the Japanese main island (60-70 ppb) with respect to higher ones over the
oceans (>80 ppb).
The uniqueness of the performance of IASI+GOME2 to retrieve near-surface ozone is put in
evidence by comparing the same in situ measurements with other satellite retrievals, such as a
single-band IASI retrieval (described in section 2.2). We use the LISA IASI product that


offers the highest sensitivity to ozone below 6 km among three French IASI products (Dufour
et al., 2012) and also largely higher than a GOME-2 only product (shown by Cuesta et al.,
2013). The IASI LMT retrieval sensitivity peaks approximately around 3 km asl over land
and 4-5 km over ocean, thus 1 km higher than that for IASI+GOME2 both over land and
ocean (see Figures 3c-d). Figure 2b shows that the IASI only retrieval is unable to clearly
capture the high ozone concentrations observed at the surface, particularly those above 60
ppb. The scatter of points for IASI retrievals is rather flat, putting in evidence a lack of
sensitivity to LMT ozone also shown in the horizontal map of Figure 3b. The single-band
retrieval variability is much lower than that measured at the surface (the ratio of standard
deviations is 0.65 at most). Mean RMS differences are above 30 ppb and the correlation
coefficient below 0.5. On the contrary, only IASI+GOME2 does capture surface ozone
variations over whole range from 40 to 90 ppb (Fig. 2a) and shows a unique performance to
capture surface ozone variability.
**2.2  Other satellite observations**
In order to analyse the origin and evolution of ozone pollution plumes, the following
correlative datasets are used: CO and $O_3$ retrievals from IASI and $NO_2$ and $CH_2O$
observations derived from GOME-2 and OMI. Morning time (around 9h30 LT) datasets from
IASI and GOME-2 are derived from the same spectra as those used in synergism by
IASI+GOME2. OMI overpass occurs in the early afternoon (near 13h30 LT).
The CO retrievals used in the present paper are derived from IASI radiances using the FORLI
algorithm (Hurtmans et al., 2012), from the Université Libre de Bruxelles (ULB) and the
Laboratoire Atmosphères, Millieux, Observations Spatiales (LATMOS). This approach uses
pre-calculated lookup tables of absorbance cross-sections at various pressures and
temperatures, and the optimal estimation for the inverse scheme. The algorithm derives
vertical profiles of CO, on a grid of 18 equidistant layers of 1 km of depth from the surface up
to 18 km, and a unique layer from 18 to 60 km. Radiative transfer calculations use operational
MetOp-A L2 temperature and humidity profiles, surface emissivity climatologies (Zhou et al.,
2011). A priori CO profiles are taken from MOZAIC, ACE-FTS for higher altitudes
(Clerbaux et al., 2005) and the LMDz-INCA global chemistry-transport model (Haugustaine
et al., 2004). FORLI provides vertical profiles, total and partial columns of CO derived by
profile integrations, averaging kernels, error estimations and quality flags (supplied by AERIS
and LATMOS). Comparisons of CO total columns derived from FORLI-IASI showed an



agreement better than 7 % and no significant bias with respect to other satellite products (for
the northern hemisphere, George et al. 2009) and ground-based retrievals from 6 NDACC
stations (Kerzenmacher et al., 2012). A validation of lower (surface-480 hPa) and upper (480-
225 hPa) tropospheric columns with respect to MOZAIC measurements found an agreement
of respectively 21% and 10 %, and correlations of respectively ~0.8 and ~0.7 (De Wachter et
al., 2012).
In the present study, we use CO retrievals at the lower troposphere (LT) integrated from the
surface up to 6 km asl (equivalent to surface-480 hPa), validated by De Wachter et al., (2012)
and presenting heights of maximum of sensitivity located at 3-5 km of altitude (i.e. thus at the
middle of this partial column, see section 4). For estimating the ratio of enhancements of $O_3$
and CO at the LMT (hereafter referred to $\Delta O_3/\Delta CO$) during individual long-range transport
events, we use $O_3$ and CO satellite observations (mixing ratios in ppb) after subtracting
background levels, as done for analysing airborne in situ data during the PHOBEA I and II
experiments over northeast Pacific (Price et al., 2004). We empirically estimate these
background concentrations as the daily average concentration minus the standard deviation
over the region of analysis (20-48°N 110-150°E). For the event in early May 2009, we derive
background levels around ~46 ppb and ~126 ppb for the observations of respectively LMT $O_3$
and LT CO. We use the same criteria for deriving $\Delta O_3/\Delta CO$ at the LMT from models (WRF-
Chem and CHASER, see sections 2.3 and 4).
We also use single-band IASI only retrievals of ozone (Eremenko et al., 2008; LISA product
in Dufour et al., 2012) in order to analyse the distribution of ozone between 3 to 6 km asl.
This single-band IASI product is similar to IASI+GOME2, but only using infrared
measurements (both approaches using a Tikhonov-Philips regularization and the KOPRA
radiative transfer code in similar configuration). By comparing IASI only and IASI+GOME2
ozone retrievals, one may identify ozone plumes located below 3 km asl and those located
between 3 and 6 km asl (as remarked by Cuesta et al., 2013). Indeed, we expect that high
ozone concentrations clearly depicted by IASI+GOME2 and not by IASI are located at the
LMT below 3 km asl. Ozone plumes located at 3-6 km asl are shown by both IASI+GOME2
and IASI. Note that since DOF for IASI+GOME2 is lower than 1 at the LMT, multispectral
outputs depend as well on ozone concentrations up to 5 or 6 km asl and they alone cannot tell
whether the ozone plumes are located in the LMT or at 3-6 km asl.





Satellite retrievals of $NO_2$ and $CH_2O$ from GOME-2 (overpass around 9h30 LT) and OMI
(13h30 LT) are used to indicate the availability of ozone precursors. These datasets are
provided by TEMIS (http://www.temis.nl) and BIRA-IASB for $CH_2O$
(http://h2co.aeronomie.be). Retrievals of $NO_2$ from GOME-2 and OMI are derived
respectively by the algorithms TM4NO2A version 2.3 (Boersma et al., 2004) and DOMINO
version 2.0 (Boersma et al., 2011). These approaches follow 3 steps: using Differential
Optical Absorption Spectroscopy (DOAS) to obtain $NO_2$ slant columns from reflectance
spectra, separating the stratospheric and tropospheric contribution to the slant column and
converting the tropospheric slant column to a vertical column with the tropospheric air mass
factor. Uncertainties for tropospheric $NO_2$ retrievals are estimated as 35-60 % for GOME-2
and 25 % for OMI. Total columns of $CH_2O$ are retrieved with the BIRA-IASB algorithm
version 14 (De Smedt et al., 2008), also based on DOAS technique and air mass factor
estimations. Uncertainties of single $CH_2O$ slant column observations typically range from
about 10–200 %, when exceeding the global background (~4 $10^{15}$ molecules $cm^{-2}$). For
reducing noise, datasets are averaged in regular grids of 1 x 1° and 2 x 2° respectively for
$NO_2$ and $CH_2O$.
### 2.3  Chemistry-transport models: WRF-Chem & CHASER
In the present study, we use WRF-Chem and CHASER chemistry-transport models for
completing the description of the ozone pollution outbreak across East Asia in early May
2009 and for verifying consistency with satellite observations. While WRF-Chem is a
regional model (Grell et al. 2005) operating in forecast mode over Asia (Takigawa et al.,
2007), CHASER is a global model (Sudo et al., 2002) with coarser spatial resolution but with
enhanced accuracy through assimilation of several satellite retrievals of atmospheric
pollutants (Miyazaki et al., 2012, 2015) and with higher model top height. These models
provide useful insights on the detailed vertical distribution (e.g. surface concentrations) and
diurnal evolution of tropospheric $O_3$ and CO over East Asia, in complement to daily satellite
observations. The daily evolution of ozone enhancement in the LMT is compared between the
models and the satellite data. For this, we estimate background levels of $O_3$ and CO for each
dataset using the same criteria (i.e. daily average minus standard deviation over the domain,
see section 2.2). We derive concentrations of $O_3$ and CO at the LMT by vertical integration
from the surface up to 3 km asl (without any smoothing by the satellite AVKs). Moreover, we
perform sensitivity studies with CHASER (in forecast mode), accounting or not for the



stratospheric contribution of ozone in order to identify the tropospheric or stratospheric origin
of this pollutant.
For the current application, WRF-Chem resolution is set to 37 vertical layers from the surface
up to 100 hPa (~15 km of altitude) and a Lambert conformal conic projection of horizontal
pixels of about ~0.9°x0.9°. Emissions from automobiles and other anthropogenic sources are
taken from EAgrid 2000 (East Asian Air Pollutant Emissions Grid Inventory; K. Murano,
personal communication) and the JCAP (Japan Clean Air Program; Kannari et al. 2007) over
Japan. Surface emissions over China and North and South Korea are taken from REAS
(Regional Emission Inventory in Asia; Ohara et al. 2007), and over Russia from EDGAR
(Emission Database for Global Atmospheric Research; Olivier et al. 1996). Biogenic
emissions are based on Guenther et al. (1993). The lateral boundaries of chemical species are
taken from the global CHASER model (Hasumi et al. 2004) every 3 hours. The system is
driven by meteorological data from the mesoscale model (MSM) of the Japan Meteorological
Agency (JMA).
CHASER analyses are obtained from an advanced chemical data assimilation system
combining satellite observations of several chemical compounds: $NO_2$, $O_3$, CO, and $HNO_3$
measurements from OMI, Tropospheric Emission Spectrometer (TES), Measurement of
Pollution in the Troposphere (MOPITT), and Microwave Limb Sounder (MLS). Assimilation
is performed according to the local ensemble transform Kalman filter technique (Hunt et al.,
2007), which simultaneously optimizes targeted chemical species, as well as the emissions of
$O_3$ precursors (i.e., NOx and CO), while taking their chemical feedbacks into account.
Comparisons against independent data show that the data assimilation results in substantial
improvements (reduced biases for tropospheric $NO_2$ by 40–85 %, for CO in the Northern
Hemisphere by 40–90 % and for $O_3$ bias in the middle and upper troposphere from 30–40 %
to within 10 %). The CHASER forecast model includes detailed chemical and transport
processes in the troposphere, including 88 chemical and 25 photolytic reactions with 47
chemical species, and has a horizontal resolution of T42 (2.8°) and 32 vertical levels from the
surface to 4 hPa. CHASER is coupled to the atmospheric general circulation model, Center
for Climate System Research/National Institute for Environmental Studies (CCSR/NIES)
atmospheric general circulation model (AGCM) ver. 5.7b. Anthropogenic, biomass burning
and biogenic/soil emissions are respectively based EDGAR version 3.2 (Olivier et al., 2005),
the Global Fire Emissions Data base (GFED) version 2.1 (Randerson et al., 2007) and the



Global Emissions Inventory Activity (GEIA) inventory (Guenther et al., 1995). Surface
emissions over Asia were obtained from REAS. For the sensitivity analysis, CHASER
simulations of ozone "from the troposphere" (not accounting for stratospheric ozone) are
obtained by setting to zero ozone concentrations above 100 hPa, starting from 1 January 2009.
Differences between ozone concentrations from full simulations and those not accounting for
the stratospheric contribution provide an estimation of the distribution of ozone transported
"from the stratosphere".
**2.4   Meteorological data: ERA-Interim reanalyses & Hysplit dispersion model**
Meteorological conditions leading to production of ozone pollution and transport across East
Asia are described in section 3 with ERA-Interim reanalyses (Dee et al., 2011) produced by
ECMWF (European Centre Medium Weather Forecast). We use meteorological fields
(downloaded from http://climserv.ipsl.polytechnique.fr) with global coverage, a horizontal
resolution of $0.75° \times 0.75°$, 37 pressure levels, and a time step of 6 h (interpolated for other
hours). Wind, geopotential height, and equivalent potential temperature fields describe
atmospheric circulation and the locations of synoptic high and low-pressure systems.
Additionally, forecasted atmospheric boundary mixing layer top heights from ERAI are used
in the analysis of the vertical distribution of LMT ozone derived from IASI+GOME2.
Trajectories of polluted air masses transported across East Asia are estimated using the
HYSPLIT dispersion model (Stein et al., 2015; Rolph et al., 2017, https://ready.arl.noaa.gov).
This tool can simultaneously track a total of 12500 air parcels released at a location and
altitude and transported during 24 hours according to meteorological fields. For the present
analysis, we set the starting altitudes from the surface up to 3 km (i.e. at the LMT which is the
layer observed by IASI+GOME2) and use built-in model reanalysis from NCEP/NCAR
(National Centers for Atmospheric Prediction / National Center for Atmospheric Research)
with a horizontal resolution of $2.5° \times 2.5°$ and 18 (29) pressure (sigma) levels. We run the
dispersion model at 0 UTC (9h00 LT in Japan, close to the MetOp satellite overpass) for each
day of the ozone pollution event at the mean arrival location of the trajectories from the
previous day. The starting point for the pollution event was determined at the south of the
North China plain, as suggested by model simulations and high concentration of ozone
precursors (see section 3.1).



## 3   Ozone pollution outbreak across East Asia in early May 2009
In this section, we describe the daily evolution of a major ozone pollution outbreak initiated
over China and transported across East Asia during the period 2-9 May 2009. First, we focus
on the formation of a large ozone plume over the North China Plain (NCP, section 3.1). Next,
we analyse the transport of these polluted air masses over the NCP in the northeastern
direction (section 3.2), and their subsequent advection by an anticyclonic circulation over
Northern China and the Korean Peninsula (section 3.3). Finally, these ozone plumes split into
two filaments and reach Japan and the Pacific far from the main sources of ozone precursors
(section 3.4). According to EANET/GAW/JAMSTEC surface observations over Japanese
islands, this is one of the two largest ozone pollution outbreaks reaching Japan during the
springtime of 2009.
### 3.1   Ozone plume formation over the NCP on 2 May
The NCP is a well-known hotspot of pollutant emissions of worldwide relevance (e.g. Richter
et al., 2005). On 2 May 2009, large concentrations of ozone precursors, such as nitrogen
dioxide ($NO_2$) and formaldehyde ($CH_2O$), are observed over the NCP (see GOME-2 satellite
retrievals at 34-37°N 113-117°E marked as a magenta rectangle in Figure 4a-b). Whereas the
dense $NO_2$ plume is mainly formed over the NCP, the highest concentrations of $CH_2O$ are
mainly located south of it (at 25-36°N 112-120°E) reaching the southern part of the NCP. The
$NO_2$ and $CH_2O$ concentrations observed on 2 May 2009 are approximately a factor ~2 higher
than the regional monthly average (also estimated with GOME-2). According to MODIS
active fire data over this region, wildfires are negligible in early May 2009 (only very few fire
spots are detected, see http://firms.modaps.eosdis.nasa.gov). The short lifetimes of these
reactive gases (up to a few hours) prevent the influence of long-range transport. The observed
$NO_2$ and $CH_2O$ plumes are then likely associated with local anthropogenic emissions from
this densely populated and industrialized region. In the following (sections 3 and 4), we
particularly focus our analysis on the daily evolution of pollutant concentrations originated
from these air masses as they are transported across East Asia (magenta/red rectangles in
Figs. 4-12).
According to model simulations, ozone concentrations over the NCP are relatively low
(below 50 ppb) during the morning of 2 May, both at the surface and within the LMT (up to 3
km asl, shown in Figures 4c,e for WRF-Chem). Therefore, we do not observe any significant





residual ozone plume from the previous day within the LMT. As expected for this time of the
day (9h00 LT), the mixing boundary layer over land is rather shallow (with its top below 1
km asl, see blue isolines representing the depth of the mixing boundary layer in Fig. 4e).
Weather conditions over the region are characterized by very low windspeeds at the lower
atmospheric levels associated with marked anticyclonic conditions (see high pressures
approaching the NCP from the west at 25-35°E 110-115°E in Fig. 4c-d). Due to partial cloud
cover, IASI+GOME2 retrievals are not available over this region during this day.
Following up with the typical diurnal cycle, ozone is photo-chemically produced during the
afternoon, until reaching concentrations of ~90 ppb near the surface in the southern part of the
NCP at 16h00 LT (according to the WRF-Chem model around 34°N 115°E, see Fig. 4d). In
the afternoon, the mixing boundary layer over this region is deeply developed with its top
near ~2 km above ground level - agl (3.5 km asl, Fig. 4f), thus suggesting that the ozone
plume freshly formed during the afternoon of this day is well mixed within the LMT.
**3.2    Ozone plume transported over the NCP on 3-4 May**
In the morning of 3 May, anticyclonic conditions prevail over the NCP with a pressure
maximum over central China, at the southern outskirts of the NCP (see Fig. 5c). According to
the Hysplit dispersion model, the ozone plume formed the previous afternoon over southern
NCP is transported by weak southerly winds until 35°N 115°E (magenta rectangle in Fig. 5a-
b). At this location, high concentrations of both LMT ozone (~90 ppb) and LT carbon
monoxide (~290 ppb) are observed from space respectively by IASI+GOME2 (Fig. 5a) and
IASI (Fig. 5b). Co-localisation of both pollutant plumes over the NCP suggests that these high
ozone concentrations are associated with surface anthropogenic emissions, as CO is a tracer
for combustion-related emissions.
Meanwhile, another ozone plume is observed over northeastern China (north of 42°N and at
115-135°E in Fig. 5a). The origin of this plume is likely associated with a low-pressure
system (see "L" and concentric isobars north of 44°N and at 120-135°E in Fig. 5c). Such
systems may both entrain ozone from the stratosphere and the UTLS region down to the
lower troposphere and also mix pollution-related ozone within the low atmospheric levels, as
analysed for other events in the same region by Dufour et al. (2015). Both phenomena may
occur in this case. On one hand, downward transport of ozone from the stratosphere (likely
west of 120°E and north of 42°N) is suggested by enhanced potential vorticity at 300 hPa (a





tracer of stratospheric air masses) north of 48°N 130°E (Fig. 5d). On the other hand, an
anthropogenic contribution of LMT ozone is indicated by the presence of a CO plume
observed by IASI east of 122°E (and north of 40°N, Fig. 5b) likely originating from
northeastern Chinese emissions. This pollution plume is observed ahead of a cold front (violet
curve in Fig. 5c), associated with the low-pressure system north of 44°N. In this region, we
expect the formation of a warm conveyor belt. This ascending air stream typically mixes up
the air masses near the surface within the low atmospheric levels (e.g. Cooper et al., 2002;
Ding et al., 2009; Foret et al., 2014). Such vertical mixing likely contributes to the
observation of near-surface pollutants by satellite retrievals sensitive within the LMT. At this
location, models show ozone concentrations only enhanced up to ~50 ppb (Fig. 6b for WRF-
Chem), near background levels. CHASER clearly simulates the contribution of stratospheric
ozone down to the LMT (north of 42°N), but shifted to the west (95-105°E, not shown). As
the paper does not focus on these air masses, a detailed analysis of these differences is beyond
the scope of the current paper.
Over the NCP (and also south of it), models simulate relatively high ozone concentrations
(~70 ppb) within the LMT at 10h00 LT (see WRF-Chem in Fig. 6b), in fair agreement with
IASI+GOME2 retrievals (Fig. 5a). At the surface, ozone concentrations simulated by WRF-
Chem remain rather low (near 40 ppb) at this time of the day (Fig. 6a), probably due to
titration during the previous night. This suggests that the high ozone concentrations observed
by IASI+GOME2 at the LMT (over the NCP) likely correspond to a plume located within the
residual boundary layer and formed during the previous day. Such an ozone plume is
simulated by WRF-Chem between 1 and 3 km of altitude at 35-38°N, while reaching the
surface south of 35°N (Fig. 6c for WRF-Chem). Considering this vertical distribution of $O_3$,
we track $O_3$-enriched air masses using a top altitude of 3 km asl for air particles in the Hysplit
dispersion model. Both WRF-Chem and co-localisation with a plume of CO confirm the
anthropogenic origin of the ozone plume at the LMT observed by IASI+GOME2 over the
NCP. At this continental location, high $NO_2$ concentrations are also both observed by GOME-
2 (Figs. 6d) and simulated by the models (not shown).
The origin of this ozone plume may also be estimated from a comparison between CHASER
simulations in two configurations: accounting or not for the contribution of ozone from the
stratosphere (see more details in section 2.3). This analysis suggests that the stratospheric
contribution over the NCP is practically negligible (~5 ppb) at the LMT (Fig. 6e). Only above




the LMT (between 3 to 6 km asl), a higher contribution of ozone (~30 ppb) from a
stratospheric filament is depicted by CHASER along a front extending from 32°N 110°E to
48°N 142°E (Fig. 6f). This is also suggested by co-located high values of PV at 300 hPa from
ERAI (Fig. 5d). IASI-only retrievals also depict this ozone filament above 3 km asl (although
most pixels are cloudy, not shown).
During the following day (4 May), the anticyclone slowly moves northeastwards and
approaches the Yellow Sea, with its pressure maximum at sea level near the coast (36°N
122°E, Fig. 7c). According to Hysplit, the ozone plume located in the residual boundary layer
the previous day is advected northwards by the anticyclonic circulation up to 39°N 117°E (see
magenta dots in Fig. 7c). At this location over the northern part of the NCP, co-located
plumes of LMT ozone and CO are consistently observed respectively by IASI+GOME2 (Fig.
7a) and IASI (Fig. 7b). With respect to the previous day, the observed ozone concentrations
remain rather high (~90 ppb) while CO concentrations start dropping (~270 ppb). Over
northern China (north of 41°N), LMT ozone is transported eastwards following a low-
pressure system centred at 50°N 140°E (not shown). The stratospheric filament depicted by
potential vorticity at 300 hPa east of 119°E (Fig. 7d) is transported southeastwards, far from
the location of the ozone plumes observed at the LMT.
### 3.3   O$_3$ plumes crossing Northern China and the Korean Peninsula on 5-7 May
The anticyclone reaches the centre of the Yellow Sea (36°N 122°E) on the next day (5 May,
Fig. 8c) where it remains for two more days (until 7 May). On 5 May, the CO plume observed
by IASI shows an almost identical horizontal structure as the O$_3$ plumes seen by
IASI+GOME2, extending across the Yellow Sea coast from the northern part of the NCP until
the northern frontier of Korea. The ozone plume originating from the NCP clearly follows the
anticyclonic circulation surrounding the Yellow Sea, reaching the northern coast of the
Yellow Sea on 5 May (at 41°N 124°E, magenta rectangle in Fig. 8a). According to Hysplit,
the pollution plume splits into two filaments on 6 May (i.e. two clear pathways are depicted,
Fig. 9c), one heading south towards the Korean Peninsula Fig. 9a) as entrained by the
anticyclonic circulation around the Yellow Sea and the other one north of it is transported by
eastwards winds to Northeast China (mentioned hereafter as respectively "southern" and
"northern" filaments in magenta and dotted red rectangles respectively at 39°N 127°E and
43°N 130°E, Fig. 9a). Pollutants over the Korean Peninsula are carried southwards by





relatively strong winds associated with a high and a low-pressure system respectively to the
west and to the east from these plumes (respectively centred at 35°N 120°E and 30°N 140°E
in Fig. 9c). These co-located $O_3$ and CO plumes show a progressive decrease in concentration
with respect to the previous days (down to ~220 ppb and ~70 ppb respectively for CO and $O_3$,
Fig. 9). Such (at least partly) reduction in pollutant concentrations may be induced by
horizontal dilution along transport away from their sources.
On 7 May, the southern pollution plume over Korea elongates southwards (magenta rectangle
at 37°N 128°E in Fig. 10a-b), with an apparent decrease in CO concentrations (~200 ppb). We
remark that the $O_3$ plumes over Korea (magenta rectangle) and the northeastern Chinese coast
(dotted red rectangle) are probably located below 3 km asl as they are only clearly shown by
IASI+GOME2 (Fig. 10a) and not by the IASI only retrieval (Fig. 10e).
Over the area of the northern pollution filament (over the northeastern Chinese coast),
enhancements of $O_3$ and CO concentrations are shown respectively by IASI+GOME2 and
IASI near 42°N 125-132°E (dotted red rectangles and dots in Figs. 10a-b). This pollutant
plume is observed ahead of the strong southward winds of a cold front (seen north of 42°N
between 115-125°E, violet curve in Figs. 10b), probably transporting freshly emitted pollution
(as suggested by enhanced $NO_2$ concentrations at 43-45°N 126-132°E in Fig. 10c). These
pollutants may originate from the densely populated agglomeration around the Harbin
megacity (43-46°N 125-127°E). This enhancement of pollutant concentrations along the cold
front is also clearly simulated by WRF-Chem (Fig. 10d for CO), although located slightly
west of the plumes depicted by the satellite retrievals (Fig. 10b). To account for this
difference in the location of the plumes, the dotted red square is shifted by 4° to the west in
Fig. 10d with respect to the other panels. This freshly emitted or produced pollution likely
mixes with the aged pollution air masses originating from the NCP. Moreover, we expect the
formation of a warm conveyor belt ahead of that cold front (in violet in Fig. 10b), which
typically mixes up air masses in the low atmospheric levels (as also remarked over this region
on 3 May 2009, Fig. 5). Vertical mixing of freshly emitted pollutants within the LMT may
also explain the enhancements of $O_3$ and CO observed by the satellite approaches.
Another ozone plume is observed south of the Yellow sea on 6-7 May by IASI+GOME2 and
IASI retrievals respectively at the LMT and 3-6 km asl (Figs. 9a, 10a,e). The origin of this
plume is probably related to downward transport from the stratosphere, as suggested by a co-
located PV filament at 300 hPa (green contours in Fig. 10e). CHASER simulations suggest



that this stratospheric ozone filament does not reach the LMT (Fig. 10f) and does not affect
ozone concentrations of the tracked pollution plumes over Korea and the northeastern Chinese
coast (rectangles).

### 3.4   O₃ plumes transported over Japan and the Pacific on 8-9 May

The high-pressure system moves southwards to the East China Sea on 8 May and then
eastwards until reaching the Pacific Ocean on 9 May (centred respectively at 28°N 122°E in
Fig. 11b and 27°N 130°E in Fig. 12b). The southern polluted air masses coming from Korea
(magenta rectangle) are entrained by the southwards circulation on an axis around 130°E in-
between the high and low-pressure systems (Fig. 11a-b). They reach Southern Japan (the
Kyushu island) on 8 May (31°N 130°E in Fig. 11a) and the Pacific on 9 May (27°N 130°E in
Fig. 12a). As only captured IASI+GOME2 (Fig. 11a) and not by IASI (Fig. 11c), the
moderately high $O_3$ concentrations over the Kyushu Island are probably located below 3 km
asl. Ozone and carbon monoxide concentrations at the location depicted by Hysplit (magenta
rectangles) are rather close to the background levels (particularly for CO).
The northern $O_3$ plume coming from the northeastern Chinese coast is transported over the
Japan Sea by strong counter-clockwise winds around the low-pressure system east of Japan
on 8 May (Fig. 11a) until reaching the Pacific southeast of Japan on 9 May (Fig. 12a). This
ozone plume is likely located at the LMT, as clearly depicted by IASI+GOME2 and not by
IASI (red rectangles respectively in Figs. 11a and 11c). According to CHASER and low PV at
300 hPa, we do not expect a significant contribution of stratospheric ozone reaching the LMT
over the Japan Sea on 8 May (Fig. 11d) and over central Japan on 9 May (Fig. 12d).
Therefore, these ozone plumes at the LMT are likely associated with photochemical
production along transport from precursors emitted over land (from northeastern China on 7
May and Japan on 8 May). Simulations from WRF-Chem also suggest significant
photochemical production of LMT ozone along transport from northeastern China to central
Japan and the Pacific (see section 4).
Downward transport of stratospheric ozone occurs southeast of Japan on 8-9 May, as
suggested by the location of the PV filament at 300 hPa (Figs. 11d, 12d), which travels
eastwards (located south of the Yellow Sea on 7 May). This ozone plume originating from the
stratosphere reaches the lower troposphere above 3 km asl (as observed by both
IASI+GOME2 and IASI in Figs. 11a,c, 12a,c) but not the LMT (indicated by CHASER in





Figs. 11d and 12d). This suggests distinct origins and vertical locations for the two elongated
ozone plumes southeast of Japan observed by IASI+GOME2 on 9 May (27-40°N 132-141°E
in Fig. 12a). The one closer to Japan is associated with photochemical production and other
one with stratospheric transport (indicated by green contours of PV in Fig. 12e), respectively
located at the LMT and 3-6 km asl.
**4   Photochemical production of lowermost tropospheric ozone during**
**transport**
According to the previous section, the major ozone outbreak initiated over the NCP in the
afternoon of 2 May 2009 is transported northeastwards over Northern China; it splits into two
pollution filaments and then heads southwards until reaching southern Japan on 9 May 2009.
Detailed analyses of the transport pathways of this large pollution plume (highlighted in
magenta/dotted red rectangles in Figs. 4-12) suggest two significant contributions of ozone
precursors from the NCP (2 May) and Northeastern China (northern pollution filament on 7
May). We do not observe any significant contribution of ozone from the stratosphere co-
located with these ozone plumes nor mixing with large ozone plumes formed in other regions.
In absence of local production (which occurs on 2 May and for the northern plume on 7 May),
we expect the evolution of LMT ozone to be mainly driven by either along-transport
production (linked with the availability of ozone precursors and solar insolation) or dilution of
the air masses (due to horizontal wind divergence and/or vertical mixing).
Figures 13 and 14 present a quantitative analysis of the Lagrangian evolution of the air
masses travelling on 3-9 May 2009 across East Asia from the NCP to southern Japan. These
time series show the daily evolution of a given variable averaged at the location of the
highlighted major pollutant plume, as depicted by the Hysplit dispersion model
(magenta/dotted red rectangles in Figs. 4-12), each day during the morning (at the time of
overpass of the MetOp-A satellite around 9h30 LT). The key variable to analyse is the ratio
$\Delta O_3/\Delta CO$ that describes the relative production or decrease of $O_3$ with respect to CO along
transport (e.g. Price et al, 2004) and also used at given fixed locations (e.g. Chin et al., 1994).
Figures 13a-c show two curves, one corresponding to the beginning of the event and the
southern filament of pollution and the other for the northern pollution filament (i.e. curves
respectively in magenta and dotted red).



## 4.1 Southern pollution filament

For the southern filament, a sustained increase of the ratio $\Delta O_3/\Delta CO$ is clearly derived during the whole ozone outbreak across East Asia from satellite observations (Fig. 13a). It evolves from ~0.25 over the NCP on 3 May to ~0.46 on 8-9 May over the Pacific and southern Japan (in magenta in Fig. 13). As dilution of the air masses affects equally the concentrations of both pollutants, a monotonous enhancement of $\Delta O_3/\Delta CO$ clearly puts in evidence the production of $O_3$ along transport. The evolution of this ratio from 0.25 over the main source regions to 0.46 after long-range transport estimated here with satellite retrievals is consistent with other estimations from airborne in situ measurements ranging from 0.2 to 0.5 (from flights at 2-3 km of altitude) for other events of transpacific long-range transport of industrial/urban pollution and in absence of stratospheric intrusions (Price et al., 2004). Overall consistency is also found with model estimates of 0.3 for typical air masses downwind from Asian pollution sources in springtime (Mauzerall et al., 2000) and the same value from in situ ground-based measurements over the United States in summer (Chin et al., 1994). These IASI+GOME2-$O_3$/IASI-CO values are fairly higher than those estimated in the lower troposphere with IASI-$O_3$ only retrievals (0.16-0.28 for the column below 6 km asl) for an Eastern Asian pollution event in May 2008 (Dufour et al., 2015) and lower than retrievals in the free troposphere (400-700 hPa) from OMI/AIRS (~0.6) over Tokyo (Kim et al., 2013).

Assuming that most CO is emitted at the beginning of the event (for the southern filament, magenta curve in Fig. 13a), the evolution of the $\Delta O_3/\Delta CO$ values suggests a production of $O_3$ along transport of ~60 % after the first 3 days (mainly over Northern China and Korea) and of ~84 % during the whole event (6 days) with respect to that over the NCP. In this case, the greatest growth of $\Delta O_3/\Delta CO$ occurs on 3-6 May when the air masses are transported over the most industrialized areas (the NCP and Northern China) with the greatest emissions of ozone precursors as $NO_x$ (as shown in Fig. 10c for $NO_2$). In the following days, a slower growth with almost constant $\Delta O_3/\Delta CO$ occurs over the Korean Peninsula (on 7 May), southern Japan (8 May) and the Pacific (9 May), far from the main sources of $O_3$ precursors over China. Less ozone production over this oceanic region is well consistent with low availability of $NO_2$ indicated both by observations and the CHASER model (Fig. 14h) and a regime of $NO_x$-limited photochemical production of $O_3$, as observed over the Fukue Islands (Kanaya et al., 2016). This behaviour is also simulated by WRF-Chem, showing ozone diurnal cycles with greater ozone production in the afternoon of the first 3 days of the event and less significant



after (see hourly outputs of the model in Fig. 14a). Simulated diurnal cycles of ozone also
reveal the strong nocturnal reduction in ozone concentrations (down to 30-40 ppb, dotted light
blue curve in Fig. 14a) particularly significant over China (3-6 May), probably associated to
titration over the continent, and less pronounced after near or over the ocean. Moreover, the
model clearly suggests a reduction in CO concentrations (also observed by IASI in the lower
troposphere), particularly significant after 6 May (Fig. 14b) and likely linked to horizontal
dilution by atmospheric transport.
The $\Delta O_3/\Delta CO$ ratios derived from the CHASER and WRF-Chem models at LMT of altitude
(asl) follow a similar relative evolution as that from satellite retrievals, with a minimum at the
beginning of the event and a relative monotonous increase by the end (Fig. 13b-c). This is
particularly observed for WRF-Chem and until 6 May for CHASER. In absolute values, the
ratio $\Delta O_3/\Delta CO$ derived from the satellite measurements is consistent with that from models.
At the beginning of the event (3-5 May), satellite estimates of the ratio are 0.1 to 0.15 higher
than those from satellite. Afterwards, satellite and WRF-Chem ratios are rather close (with
differences between 0.05 and 0.1). Differences between the models and with respect to
satellite-derived $\Delta O_3/\Delta CO$ ratios are likely associated with photochemical schemes in the
models, model resolutions and probably to a lesser extent (as we use ratios of $\Delta O_3$ and $\Delta CO$)
with precursors availability, the location of the plumes, etc.
Figures 14e-f show that the steady increase of $\Delta O_3/\Delta CO$ observed for the southern pollution
plume does not seem to be linked to changes in sensitivities of the satellite retrievals. Neither
the degrees of freedom nor the altitude of maximum sensitivity for the analysed atmospheric
columns for either $O_3$ and CO reflect such a steady variation, greater during the first 3 days
and nearly flat afterwards, as that observed for $\Delta O_3/\Delta CO$. The LMT $O_3$ retrieval sensitivity
peaks between 2.5 and 3 km asl for most of the days (and near 4 km asl over oceanic cold
waters on 9 May), with degrees of freedom fluctuating from 0.2 to 0.3 (for the LMT and
around 5.5 to the $O_3$ total column). The CO lower tropospheric column is retrieved with 0.8 to
1 degrees of freedom with a peak of sensitivity from 3.5 to 5 km of altitude.
Figure 14 (c-d, g) show evidence of the negligible influence of stratospheric ozone on the
evolution of $\Delta O_3/\Delta CO$ at the LMT for the polluted air masses tracked in 3-9. Ozone amounts
within the LMT originating from the troposphere are a factor ~12 greater than the
contribution from the stratosphere, according to CHASER simulations (accounting or not for
stratospheric contributions, Fig. 14c). On the other hand, the ozone contribution of





stratospheric downward transport at the upper troposphere (from 6 to 12 km asl) remains
rather constant during the whole event (Fig. 14d). This is consistent with meteorological
tracers of stratospheric air masses, as the potential vorticity (Fig. 14g). At 500 hPa, no
particular enhancement of potential vorticity is clearly remarked in correlation with the days
of high concentration of ozone at the LMT. Particularly, potential vorticity on 3 May 2009 is
high only at 300 hPa in consistency with an ozone enhancement at the upper troposphere (Fig.
14d), but not below (see potential vorticity at 500 hPa in Fig. 14g). As quality check, we also
remark fair consistency between IASI+GOME2 and CHASER ozone partial columns (adding
contributions from the Troposphere and Stratosphere in Fig. 14c-d) in average over the whole
period, both within the LMT and the upper troposphere.
**4.2  Northern pollution filament**
For the northern pollution plume, satellite-derived $\Delta O_3/\Delta CO$ ratios show an increase on 6
May (curve red in Fig. 13a) with respect to the previous days, as remarked for the southern
plume. On 7 May, the eastern plume airmasses exhibit lower $\Delta O_3/\Delta CO$ ratios of ~0.25,
probably due to mixing with freshly emitted pollutants from the Northern China megacities
(suggested by CO observations on 7 May, Fig. 10b, and $NO_2$ concentrations from CHASER
in Fig. 14h). This value of $\Delta O_3/\Delta CO$ is practically the same as the one observed on 3 May
over large pollution sources from NCP. From 7 to 9 May, the $\Delta O_3/\Delta CO$ ratio (in red) raises
up monotonically from ~0.25 to ~0.4, thus suggesting photochemical production along
transport (as remarked for the days followed emission of ozone precursors over the NCP).
This evolution in terms of $\Delta O_3/\Delta CO$ ratios corresponds to an ozone production of about ~60%
with respect to that on 7 May, within 2 days. The relative evolution of satellite-derived
$\Delta O_3/\Delta CO$ ratios is consistent with WRF-Chem simulations (red curve in Fig. 13b), which also
shows a relative increase from 5 to 6 May and then lower values on 7 May (with an additional
pollution plume) that rise up monotonically until 9 May. The CHASER model shows an
enhancement from 7 to 8 May, but it drops on 9 May (Fig. 13c). The latter might be linked to
the coarser resolution of this global model and a difficulty to represent such small-scale ozone
plumes.
During this period, air masses are transported from Northeastern China to the Japan Sea, then
over Central Japan and finally reaching the Pacific. Ozone precursors might originate from
Northeastern Chinese and Central Japanese megacities. CHASER analyses suggest a



relatively higher availability of $NO_2$ at the LMT for the northern pollution filament (dotted
green curve with stars in Fig. 14h) as for the southern plume (light green in Fig. 14h). This is
consistent with the greater growth of $\Delta O_3/\Delta CO$ (and therefore ozone production) from 7 to 9
May for the northern pollution plume with respect to that at the South, as estimated with
satellite retrievals (Fig. 13a). WRF-Chem simulations also suggest the occurrence of ozone
production along transport after the 7 May by a succession of marked diurnal cycles of ozone
with greater amounts in the afternoon (Fig. 14a). As compared to the period before 6 May,
ozone diurnal cycles simulated by WRF-Chem exhibit smaller amplitudes, which are likely
associated with less nighttime titration over non-continental areas. The reduction of this ozone
reservoir may also enhance the growth of $\Delta O_3/\Delta CO$ along transport.
As for the southern plume, stratospheric contribution of ozone down to the LMT at the
location of the northern pollution filament seems negligible according to CHASER
simulations (Fig. 14c) and low values of potential vorticity (Fig. 14g). Besides, satellite-
derived $\Delta O_3/\Delta CO$ ratios may be affected by changes in sensitivity for the CO IASI retrievals
that peaks at the middle troposphere on 7-8 May, instead of the lower troposphere (Fig. 14f).
Such reduction of sensitivity to lower tropospheric CO likely induces a reduction of the
retrieved amount of CO (as CO is less abundant far from the surface). This effect would
imply an overestimation of $\Delta O_3/\Delta CO$ ratio derived on 7-8 May, but which would only
emphasize the enhancement of $\Delta O_3/\Delta CO$ estimated from satellite data between 7 to 9 May.
Conclusions drawn on the occurrence of photochemical ozone production in this period are
not affected by these changes in CO retrieval sensitivity.
**5    Summary and conclusions**
We have presented a detailed study of the daily evolution of lowermost tropospheric ozone
during a major pollution outbreak across East Asia in early May 2009, by means of
IASI+GOME2 multispectral satellite observations and chemistry-transport models. This new
multispectral satellite approach offers the currently unique capacity to observe the ozone
distribution at the lowermost troposphere (below 3 km asl) with a maximum of sensitivity
down to 2 km asl over land. Comparison with respect to ozonesonde measurements show a
good performance of IASI+GOME2 to retrieve ozone at the LMT, in average for 46 locations
in all continents around the world and for all seasons (mean bias of ~3%, correlation of 0.85
and mean precision of 16%) and also particularly over East Asia (where the present analysis



is focused). Comparisons with surface in situ measurements illustrate as well the very good
performance of IASI+GOME2 to observe ozone pollution from space. Contrary to IASI
alone, IASI+GOME2 is capable of observing the spatiotemporal variability of surface ozone
during the 2 main pollution events in springtime 2009 over the Japanese Islands, with
relatively low bias (5%) and a fair correlation (0.69).
Using IASI+GOME2, we describe the transport pathways and daily evolution of the ozone
pollution outbreak in the lowermost troposphere across East Asia in early May 2009, with
unprecedented observational detail. We document the transport pathways of lowermost
tropospheric of ozone and carbon monoxide plumes from the North China Plain to the Pacific,
surrounding the Yellow Sea and passing over Korea and Japan. Model simulations suggest
that these plumes are formed near the surface on 2 May, mixed within the mixing boundary
layer over the lowermost troposphere (up to 3 km asl) during the day and then transported as a
residual boundary layer in the following days until the Pacific on 9 May. Satellite retrievals
depict clearly concomitant structures of LMT $O_3$ and CO plumes mostly every day, thus
suggesting the anthropogenic origin of both pollutants. Within the pollution plumes, LMT $O_3$
mixing ratios range from ~90 ppb at the beginning of the event to ~70 ppb at the end. During
the event, ozone concentration is affected simultaneously by both photochemical production
within transported air-parcels and horizontal/vertical dilution associated with atmospheric
circulation. We estimate that the contribution of photochemical production is an increase of
up to 84 % of ozone amounts with respect to that produced on the first day of the event over
NCP. This estimation uses $O_3$ to CO enhancements ratios with respect to background, for the
pollution plumes transported across East Asia. The evolution of this ratio is influenced by
sources or sinks of pollutants and not by atmospheric dilution, as this last one affects equally
both pollutants. This type of results represents a strong benchmark for atmospheric pollution
models. It has been shown, that the two models used here (CHASER, and WRF-Chem) are
able to reproduce the broad features of the temporal evolution of the enhancement ratio.
Absence of stratospheric ozone contributions confirms the photochemical origin of $O_3$
enhancements with respect to those of CO. Moreover, detailed tracking of pollution plumes
suggests that it splits into two pollution filaments when crossing over Northeastern China.
One of them is mixed with freshly emitted pollutants, with significant photochemical
production of ozone, but the other one follows a rather constant evolution of the $O_3$ to CO
enhancements ratio until reaching the Pacific.



The present satellite based approach has shown original and air-quality relevant skills to
describe the evolution of transboundary pollution outbreaks across East Asia. Particularly,
distinguishing photochemical production along transport to that originally produced over
major pollution sources is a significant contribution for a better understanding of air quality
degradation and developing efficient pollution mitigation policies. Future studies will extend
the approach to longer time periods and consider multiple meteorological regimes propitious
for East Asian pollution.
**Acknowledgements**
Authors are grateful for the essential support of the Sakura Hubert Curien partnership (PHC)
for this French-Japanese cooperative study of ozone pollution over East Asia. This program is
supported by Japan Society for the Promotion of Science (JSPS) in Japan and the Ministries
of Affaires Etrangères et du Développement International (MAEDI) et de l'Education
Nationale de l'Enseignement Supérieur et de la Recherche (MENESR), and the French
Embassy in Japan. We thank the financial support of Centre National des Etudes Spatiales
(CNES, the French Space Agency) via the "SURVEYOZON" project from TOSCA (Terre
Ocean Surface Continentale Atmosphère), the Programme National de Télédétection Spatiale
(PNTS, www.insu.cnrs.fr/pnts, grant PNTS-2013-05, project "SYNAEROZON"), the
PolEASIA project (ANR-15-CE04-0005) from the Agence Nationale de la Recherche (ANR),
the Université Paris Est Créteil (UPEC), the Centre National des Recherches Scientifiques-
Institut National des Sciences de l'Univers (CNRS-INSU), for achieving this research work
and its publication.
We warmly acknowledge as well all datasets provided for this study: CO satellite retrievals
from IASI from ULB/LATMOS (Université Libre de Bruxelles/Laboratoire Atmosphères
Milieux Observations Spatiales) laboratories, particularly to C. Clerbaux et J. Hadji-Lazaro,
the French atmospheric datacentre AERIS (www.aeris-data.fr) for providing IASI data and
supporting the production of IASI+GOME2, tropospheric $NO_2$ column data and $CH_2O$ from
the GOME-2 and OMI sensors respectively from TEMIS (www.temis.nl) and BIRA-IASB
(h2co.aeronomie.be), GOME-2 level 1 data from EUMETSAT (provided by the NOAA
CLASS data portal), WRF-CMAQ simulations from Prof. K. Yamaji from the University of
Kobe, ozonesondes data from WOUDC/SHADOZ/GMD (World Ozone and Ultraviolet Data
Centre/Southern Hemisphere Additional Ozonesondes/Global Monitoring Division) networks





and surface in situ measurements of ozone from the GAW/EANET (Global Atmosphere
Watch/Acid Deposition Monitoring Network in East Asia) networks and ECMWF for
meteorological reanalysis (ESPRI climserv center for providing access to data). IASI is a joint
mission of EUMETSAT and CNES. The authors gratefully acknowledge the NOAA Air
Resources Laboratory (ARL) for the provision of the HYSPLIT transport and dispersion
model and/or READY website (http://www.ready.noaa.gov) used in this publication. We
acknowledge the Institut für Meteorologie und Klimaforschung (Germany) and RT Solutions
(USA) for licences to use respectively the KOPRA and VLIDORT radiative transfer models.
We also thank Z. Cai from the Chinese Academy of Sciences (China) and X. Liu from
Harvard-Smithsonian (USA) for their support to produce IASI+GOME2 data and fruitful
discussions, and C. Caumont from LISA for contributing to the validation of IASI+GOME2
data.

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



**Table 1.** Validation of IASI+GOME2 ozone retrievals at the LMT against ozonesondes
measurements from 46 stations distributed worldwide (and over East Asia, i.e. 3 Japanese
stations) launched in 2009 and 2010, during all seasons. We account for the satellite retrieval
sensitivity by smoothing ozonesonde profiles with averaging kernels of IASI+GOME2 co-
located pixels within +/- 1° of latitude and longitude. Ozonesonde-derived LMT ozone
columns are calculated by vertical integration and compared with the averaged of
IASI+GOME2 co-located retrievals. Biases and RMS differences are given in Dobson Units
(DU) and percentage in parenthesis. Scatterplots of these datasets are provided in Fig. 1.

| IASI+GOME2 retrievals at the LMT vs. ozonesondes | Ozonesondes distributed worldwide | Ozonesondes over East Asia |
|---|---|---|
| Bias | -0.31 (-3.1%) | 0.37 (3.3%) |
| Correlation | 0.85 | 0.76 |
| RMS difference | 1.62 (16%) | 1.43 (13%) |
| Standard deviation ratio | 1.01 | 1.00 |
| Number of ozonesondes | 1035 | 112 |





**Table 2.** Comparison of ozone in situ measurements at the surface from 11 EANET stations
over the Japanese islands with IASI+GOME2 and IASI only retrievals at the LMT, for two
major ozone outbreaks on 4-9 April and 4-9 May 2009. We consider in situ measurements at
10h00 LT and the average of co-located satellite retrievals +/- 1° of latitude and longitude.
We only account for coincidences with both IASI+GOME2 and IASI retrievals. Biases and
RMS differences are given in ppb mixing ratio and percentage in parenthesis. Scatterplots of
these datasets are provided in Fig. 2. A selection of the data with limited gradient (lower than
20 ppb in absolute value) of ozone between the surface and 2 km (according to CHASER
analysis) is considered.

| | IASI+GOME2 vs. Surface measurements | | IASI vs. Surface measurements | |
|---|---|---|---|---|
| | Limited gradient surface-2km | All cases | Limited gradient surface-2km | All cases |
| Mean bias | -3.4 (-5.4 %) | -3.0 (-4.8 %) | -15.6 (-24.7 %) | -15.6 (-24.7 %) |
| Correlation R | 0.69 | 0.63 | 0.48 | 0.46 |
| RMS difference | 12.4 (19.7 %) | 13.5 (21.3 %) | 19.5 (31.0 %) | 20.0 (31.8 %) |
| Standard deviation ratio | 1.10 | 0.97 | 0.65 | 0.57 |
| Number of measurements | 44 | 52 | 44 | 52 |





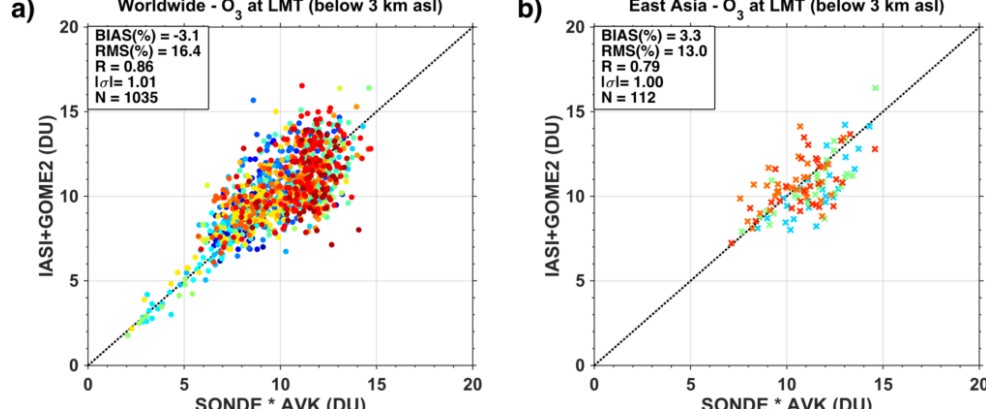

**Figure 1.** Validation of IASI+GOME2 retrieval of $O_3$ (in Dobson Units, DU) at the LMT (between the surface and 3 km asl) by comparison with ozonesondes during 2009 and 2010 launched from **(a)** 46 stations spread worldwide and **(b)** 3 Japanese stations (Sapporo, Tateno and Naha). Averaging kernels of IASI+GOME2 are used for smoothing ozonesonde measurements for accounting for satellite retrievals sensitivity.





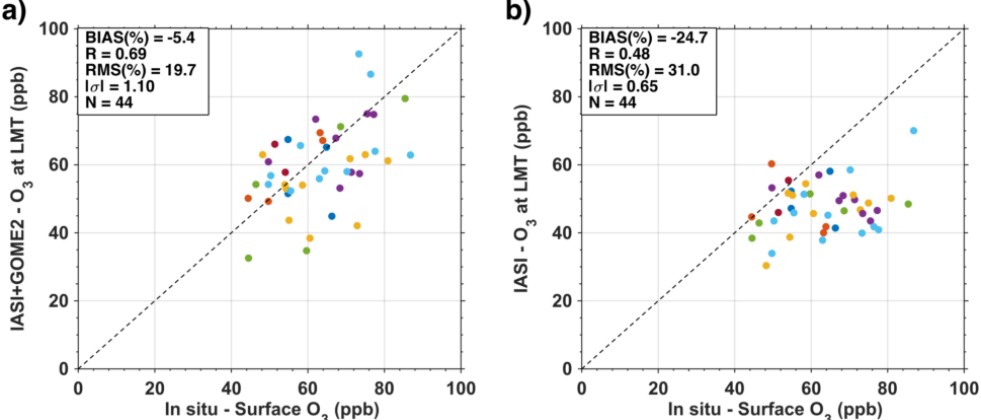

**Figure 2**. Evaluation of the capacity of IASI+GOME to retrieve near-surface ozone:
Comparisons of **(a)** IASI+GOME2 and **(b)** IASI only retrievals with surface ozone
observations from 11 EANET/GAW surface in situ stations over East Asia, during the two
greatest East Asian ozone pollution events in springtime 2009 (from 4 to 9 April and from 4
to 9 May 2009). The figures show cases with vertical gradient of ozone concentration
between the surface and 2 km of altitude below 20 ppb (according to CHASER model
analysis). This is a direct comparison without smoothing by averaging kernels. Colours
indicate different days of the comparison.



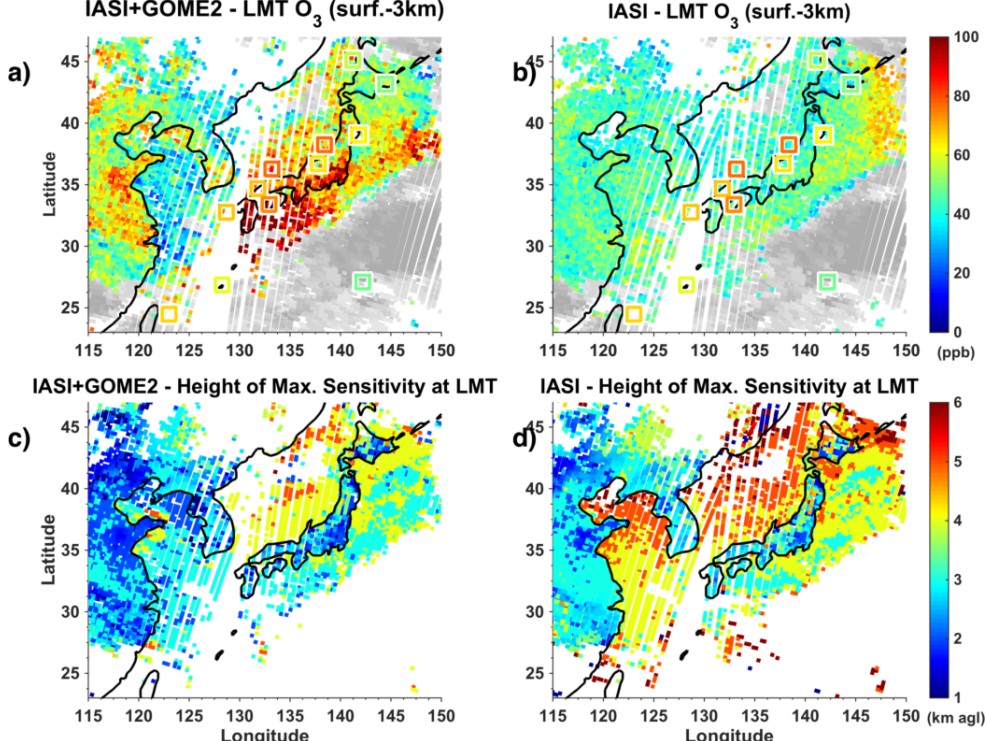

**Figure 3**. Example of comparison on 9 April 2009 over East Asia of **(a)** IASI+GOME2 and **(b)** IASI retrievals of LMT ozone (from the surface up to 3 km asl in both cases) with surface observations (squares in panels a and b). Grey-shaded pixels show cloud fractions above 0.3, as derived from GOME2 Fresco algorithm. Heights of maximum sensitivity of the LMT ozone partial columns are shown for **(c)** IASI+GOME2 and **(d)** IASI.



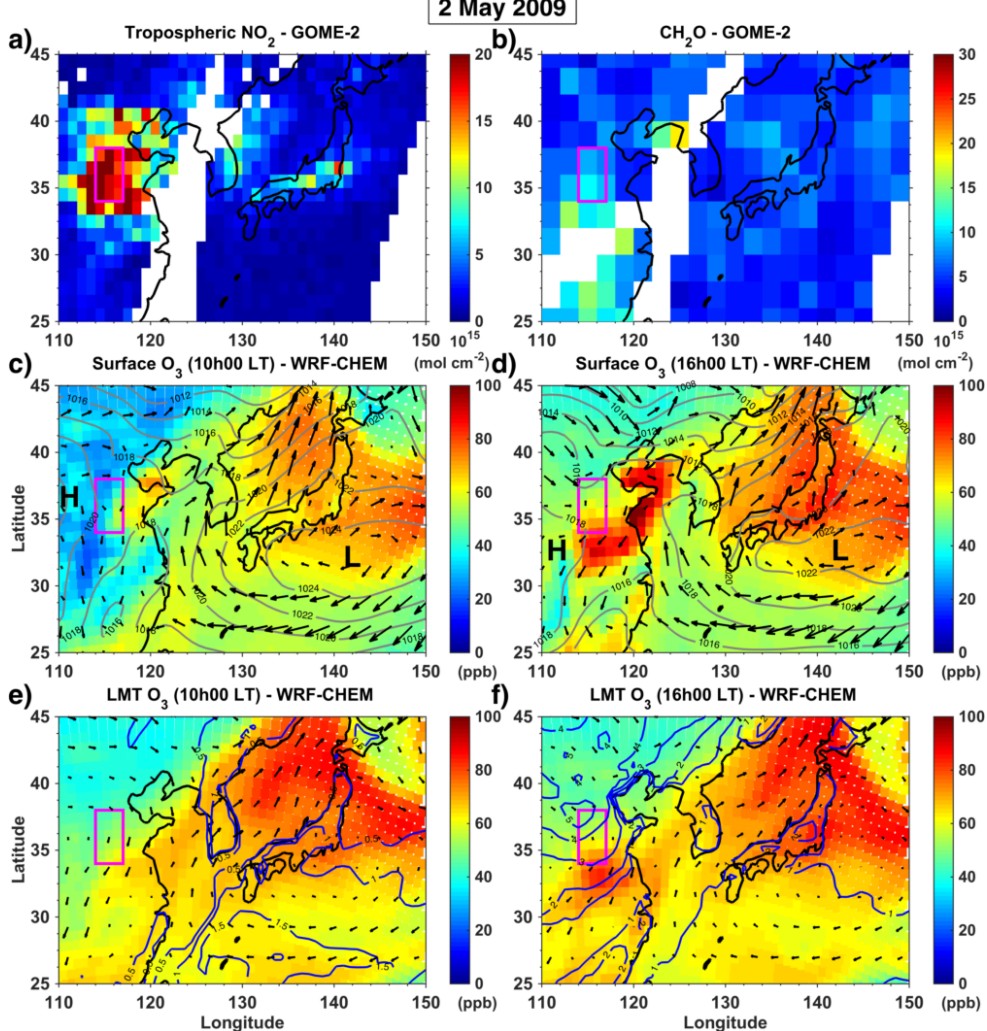

**Figure 4. (a)** Tropospheric nitrogen dioxide $NO_2$ and **(b)** Formaldehyde $CH_2O$ distribution over East Asia on 2 May 2009 derived from GOME-2 observations at 9h30 LT. **(c-f)** Ozone distribution simulated by the WRF-Chem model at the surface (panels **c** and **d**) and averaged below 3 km of altitude agl (LMT, panels **e** and **f**), in the morning (at 10h00 LT, panels **c** and **e**) and on the afternoon (at 16h00 LT, panels **d** and **f**). Iso-contours in grey (panels **c-d**) and dark blue (**e-f**) are mean sea level pressure (in hPa) and mixing boundary layer height (in km asl) from ERAI reanalysis. Arrows depict winds at the surface (c-d) and 850 hPa (e-f) from ERAI. Low and high pressure systems are indicated by respectively "L" and "H" (in panels c-d).



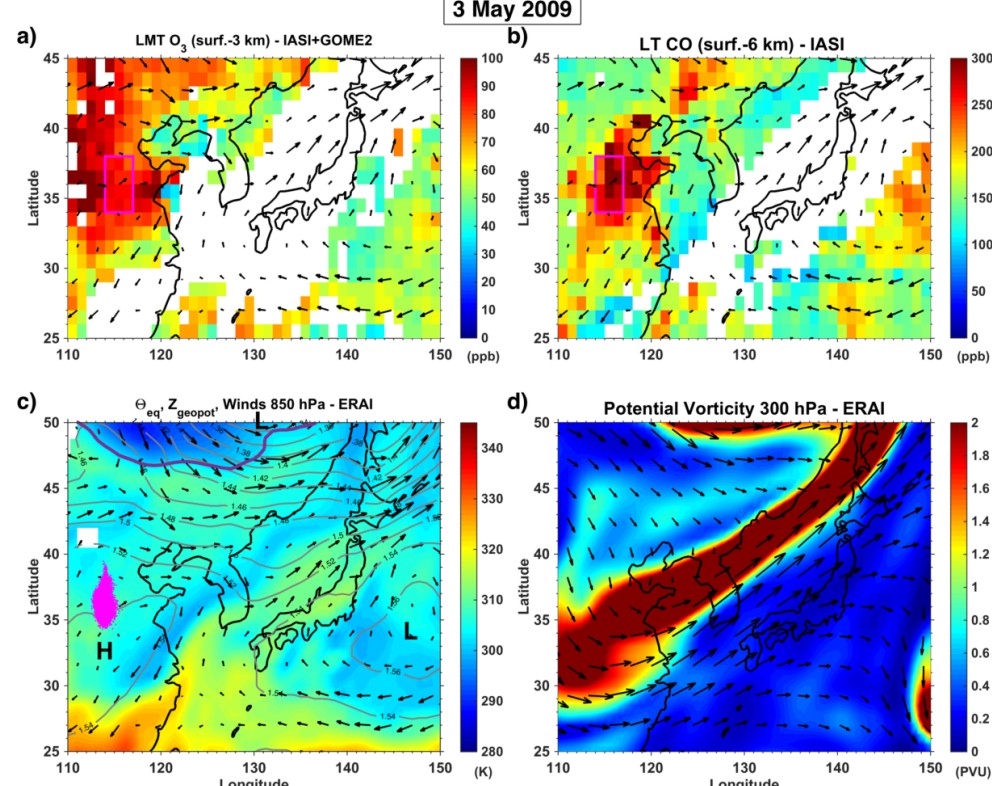

**Figure 5.** Gaseous pollutants distributions and meteorological situation over East Asia on 3 May 2009: **(a)** Lowermost tropospheric ozone derived below 3 km of altitude asl from IASI+GOME2, **(b)** Carbon monoxide at the lower troposphere (below 6 km of altitude) retrieved from IASI measurements, **(c)** Equivalent potential temperature $\theta_{eq}$ (colour shading in K), geopotential height $Z_{geopot}$ (grey isolines every 20 m), winds at 850 hPa from ERAI reanalyses, **(d)** Potential vorticity (colour shading in PVU) and winds at 300 hPa from ERAI. The magenta rectangle in panels (a) and (b) indicate the overall location on 3 May 2009 of the tracked airmasses during the ozone pollution outbreak of early May 2009 and the magenta dots in (c) correspond to the precise air parcels locations provided by Hysplit. The location of a cold front is shown by a violet curve in panel (c).



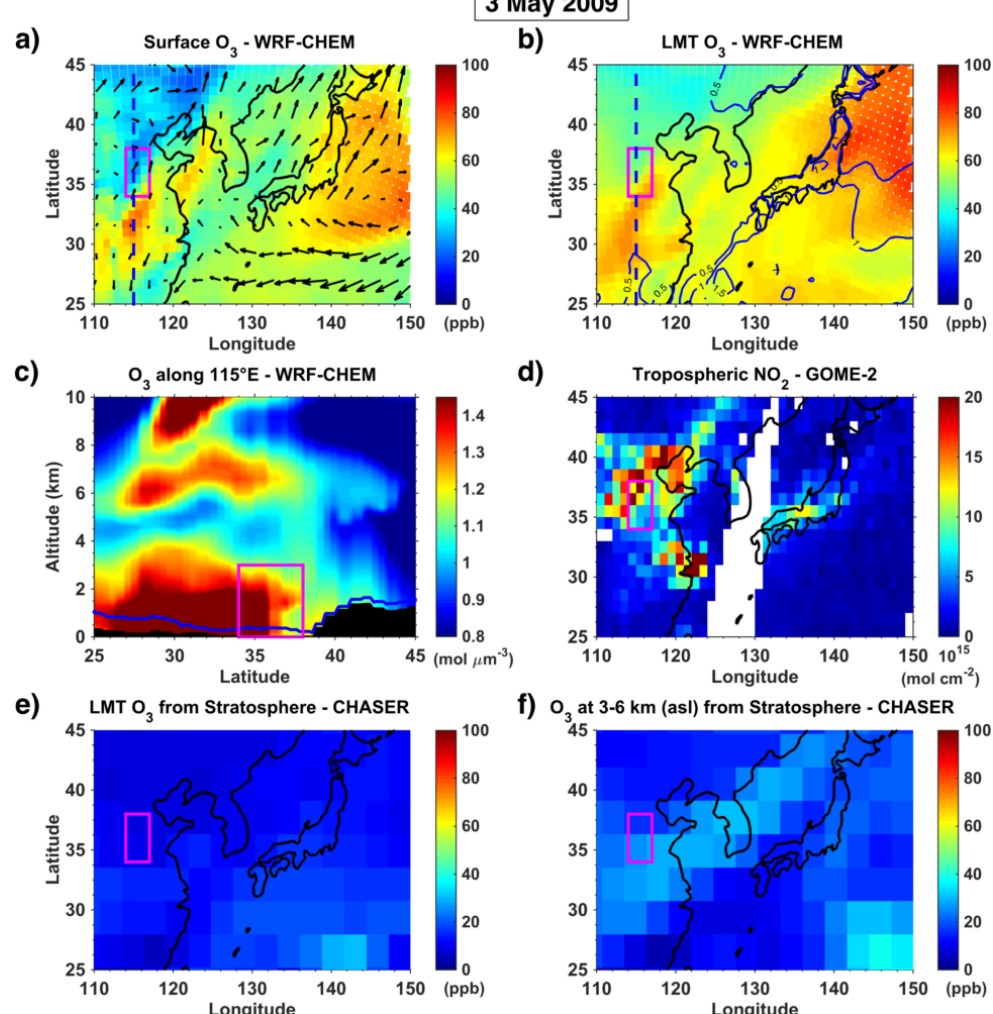

**Figure 6.** Ozone distribution over East Asia **(a)** at the surface and **(b)** at the LMT according
to the WRF-Chem model on 3 May 2009 at 10h00 LT. Panels (a) and (b) also show surface
winds (arrows) and mixing boundary layer height (blue contours), respectively. **(c)** Transect
of vertical profiles of tropospheric ozone burden (in molecules per $\mu m^3$ of air) along the axis
115°E (indicated as a dashed blue line in panel a) from derived from WRF-Chem, with the
mixing boundary layer height (blue) derived from ERAI reanalysis and orography (black
shading).     **(d)** Tropospheric $NO_2$ distribution derived from GOME-2 measurements.
Stratospheric ozone reaching **(e)** the LMT and **(f)** the atmospheric layer at 3-6 km of altitude,

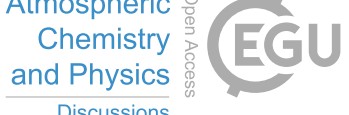



1   according to CHASER model simulations on 3 May 2009 at 10h00 LT. Magenta rectangles

2   show the locations of the air masses tracked during the pollution event in early May 2009.

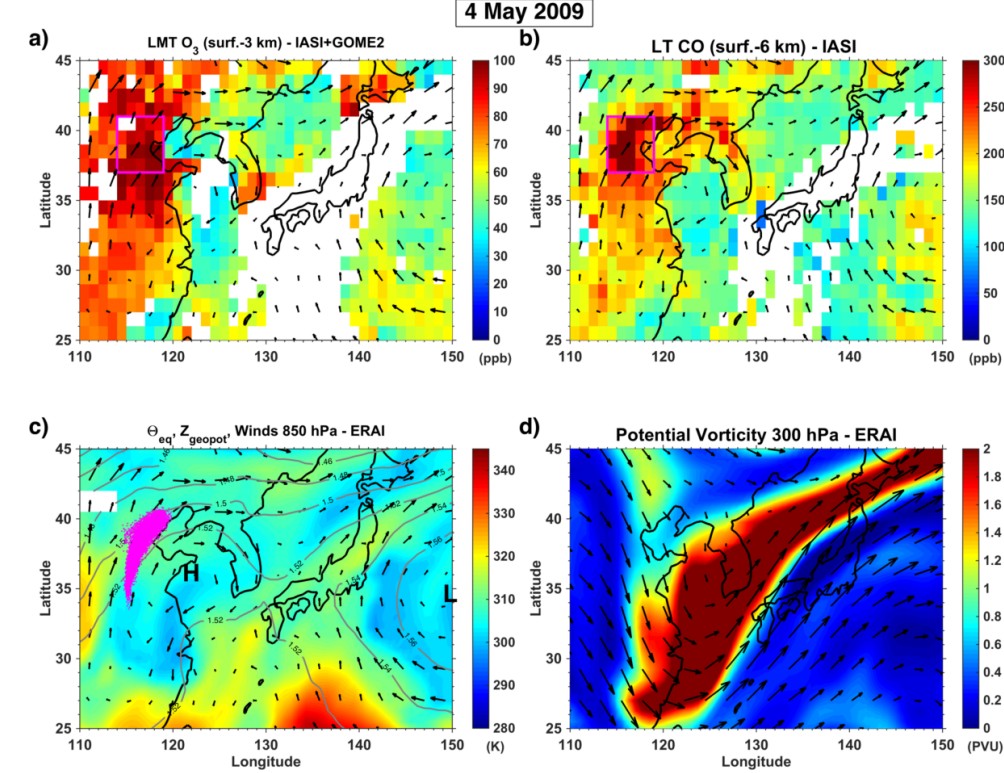

4   **Figures 7.** Idem as Fig. 5 but for 4 May 2009.



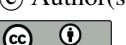

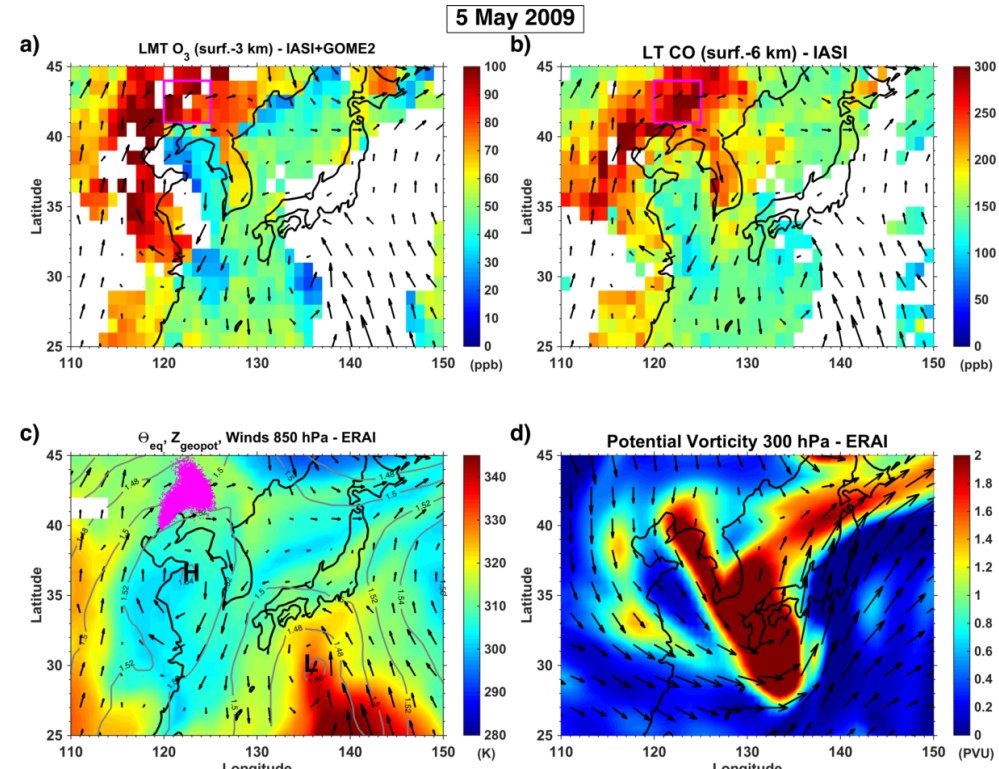

**Figure 8.** Idem as Fig. 7 but for 5 May 2009





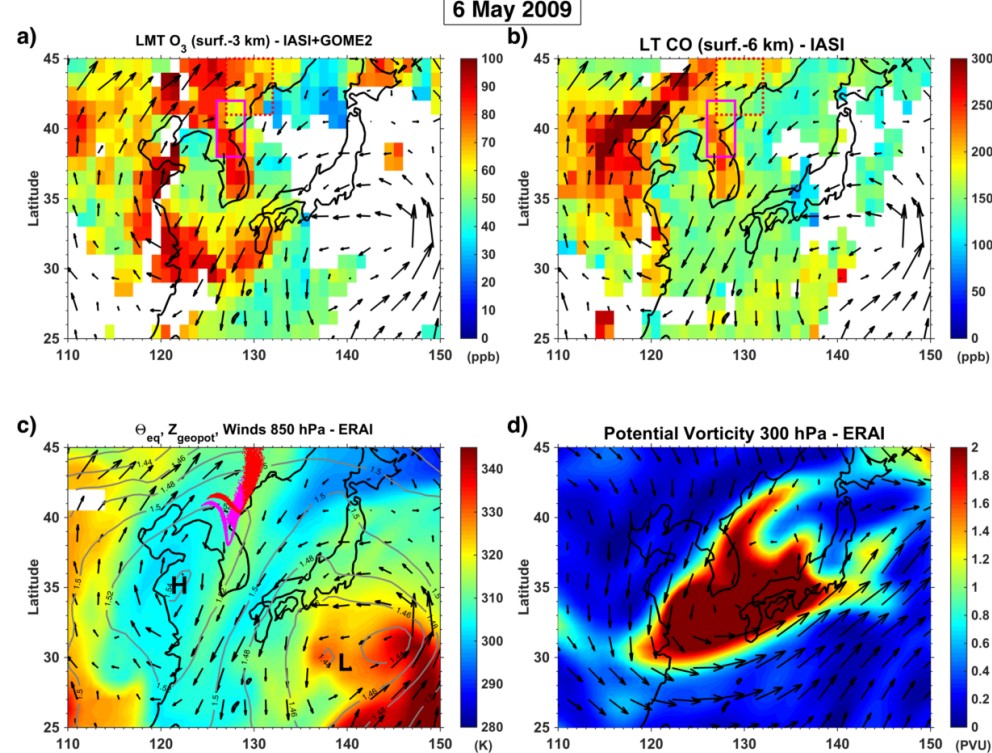

**Figure 9.** Idem as Fig. 7 but for 6 May 2009. In panels (a-b), magenta and dotted red squares
show the main location of the southern and northern pollution filaments, respectively. Dots in
panel (c) indicate the location of the air parcels tracked with the HYSPLIT dispersion model,
in magenta/red (southern/northern pollution plumes).





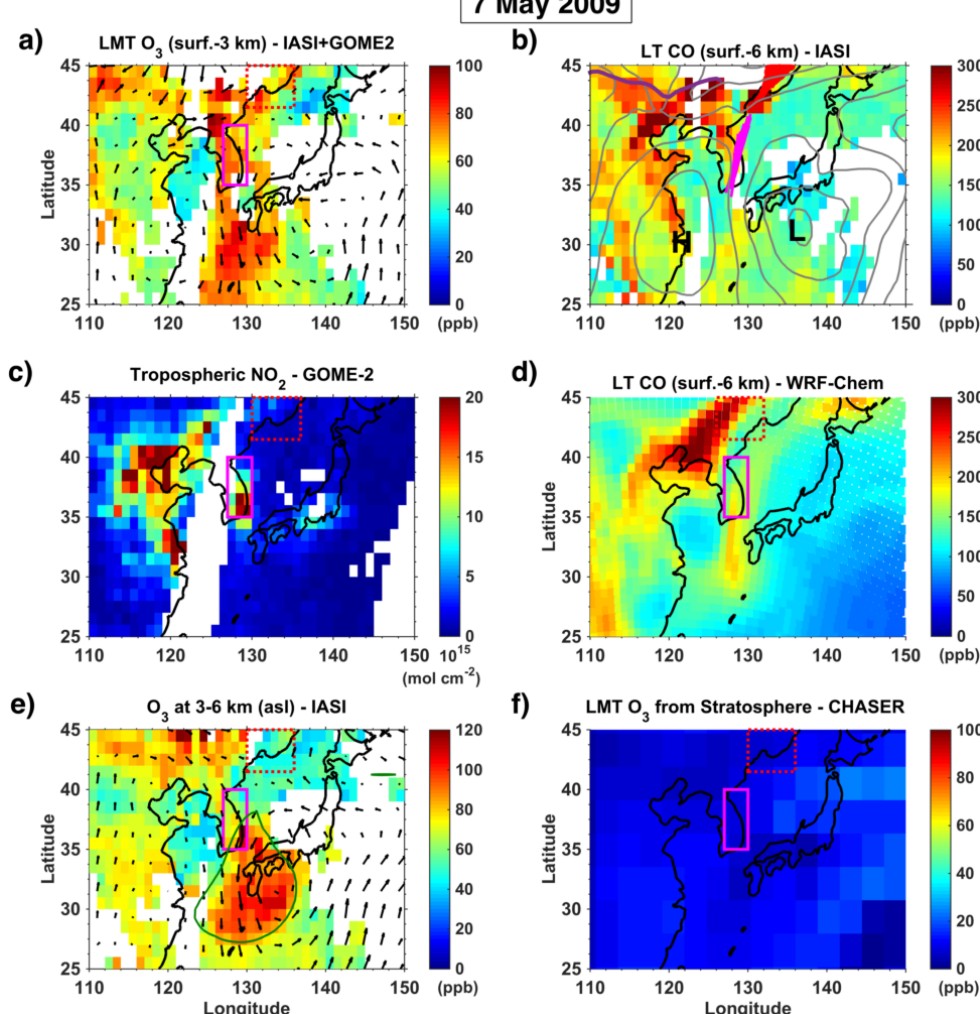

**Figures 10.** Idem as Fig. 7 for panels **(a)** and **(b)** but for 7 May 2009. **(c)** Tropospheric $NO_2$
distribution derived from GOME-2 measurements. **(d)** LT CO distribution according to WRF-
Chem model. **(e)** Tropospheric ozone from 3 to 6 km of altitude derived from IASI, winds at
700 hPa and potential vorticity contours (2 PVU in green) from ERAI. **(f)** Stratospheric ozone
reaching the LMT according to CHASER model simulations. Panel (b) also shows
geopotential heights at 850 hPa from ERAI (grey contours every 200 m) and the location for 7
May 2009 of the polluted tracked air masses derived from HYSPLIT (magenta and red dots
for the southern and northern pollution filaments). For WRF-Chem (panel d), the northern
pollution plume (dotted red square) is shifted 4° to the west, in order to account for the





1    difference in its location between the model and satellite observations. The location of a cold

2    front is shown by a violet curve in panel (b).

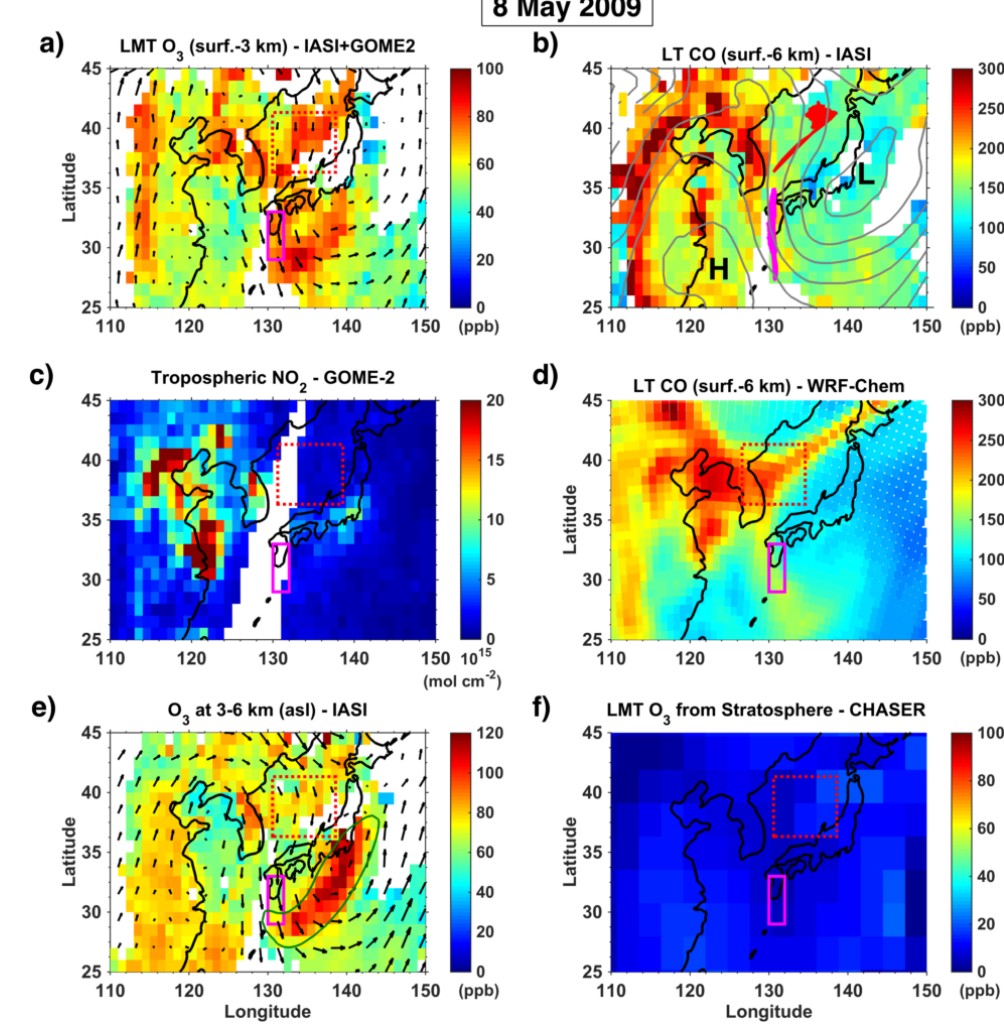

4    **Figure 11**. Idem as Fig. 10 but for 8 May 2009.



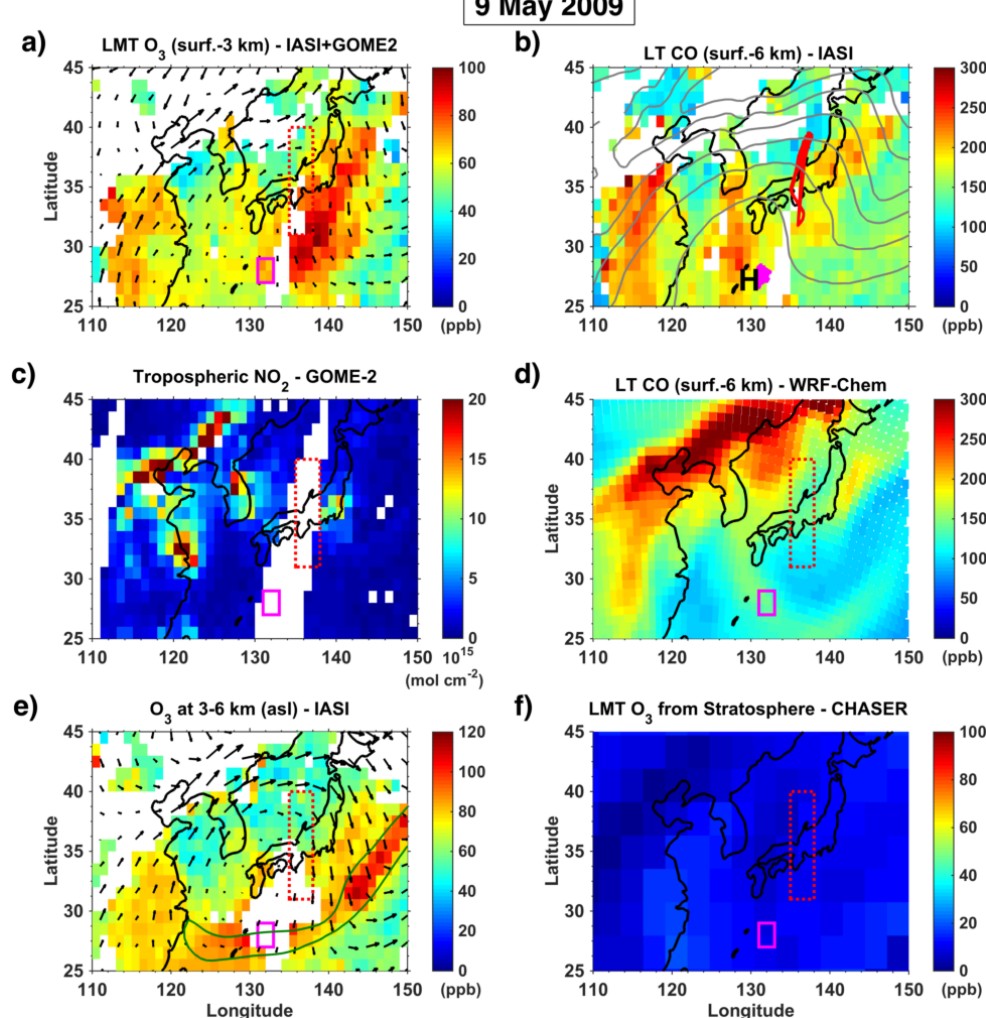

2   **Figures 12.** Idem as Fig. 10 but for 9 May 2009. Here, no shift is considered for the location

3   of northern pollution plume (dotted red square) in WRF-Chem simulations, with respect to

4   that of satellite observations.



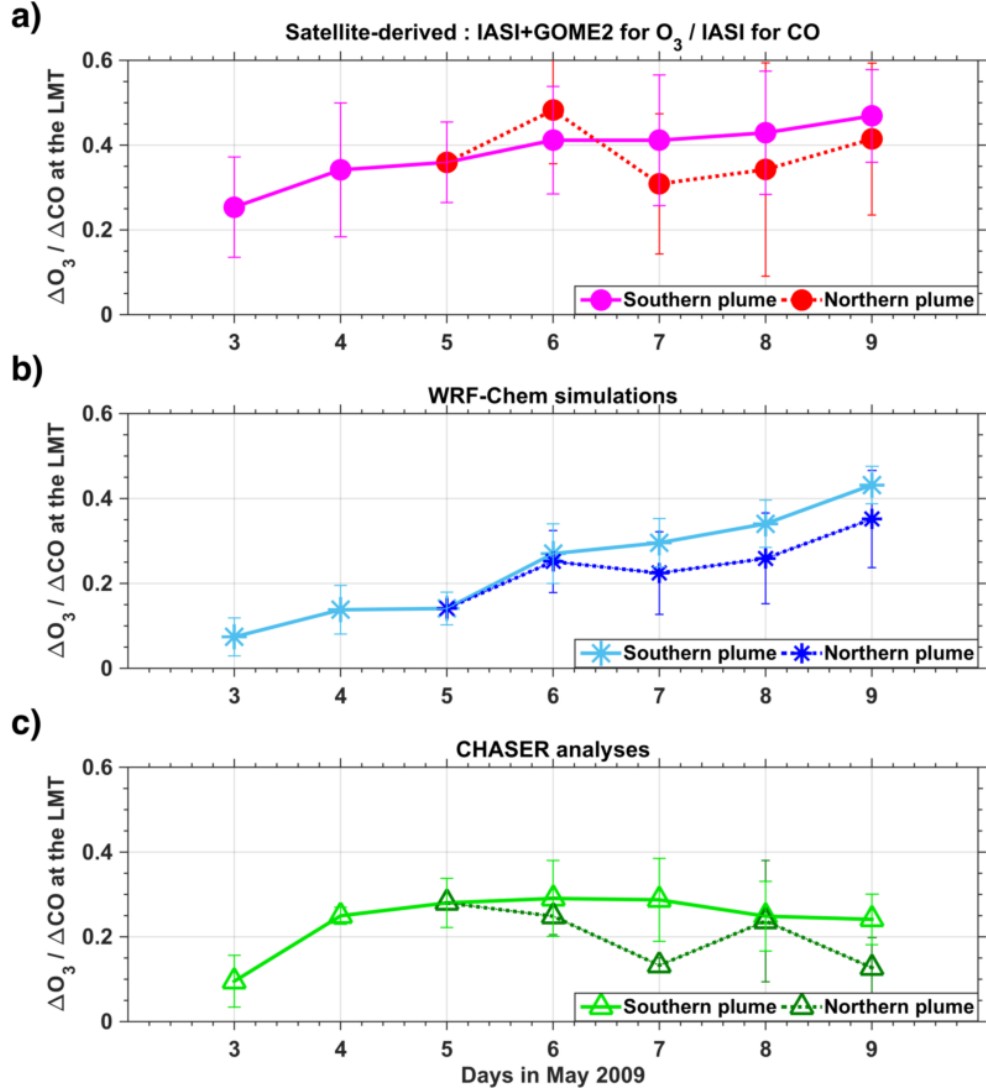

Figure 13. Lagrangian evolution of ozone enhancement along transport described by the ratio
$\Delta O_3/\Delta CO$ at the LMT for the two pollution plumes tracked across East Asia from 3 to 9 May
2009, derived from (a) IASI+GOME2 $O_3$ and IASI CO satellite retrievals, (b) WRF-Chem
simulations and (c) CHASER analyses. Ratios of $\Delta O_3/\Delta CO$ for the southern (northern)
pollution plumes are plotted in magenta (dotted red), light blue (dotted blue) and light green
(dotted green) in respectively panels (a) to (c). Curves show mean and standard deviations
(+/- vertical bars) of $\Delta O_3/\Delta CO$ over the areas depicted by rectangles in Figs. 5 and 7-11 for
each of the days of the pollution outbreak.





**Figure 14.** Lagrangian evolution at the location of the pollution plumes across East Asia (rectangles in Figs. 5 and 7-11) on 3-9 May 2009 for the following variables: **(a)** $O_3$ and **(b)** CO mixing ratios at the LMT (plain lines) and the surface (dotted lines) from WRF-Chem, for the southern (light blue) and northern (blue) pollution plumes. Ozone burden at the **(c)** LMT and the **(d)** Upper Troposphere (UT), observed by IASI+GOME2 (magenta/red for the



southern/northern plumes) and simulated by CHASER for air masses originating from the
Troposphere (squares) and Stratosphere (triangles). Satellite retrievals sensitivity in term of
**(e)** degrees of freedom and **(f)** heights of maximum sensitivity at the LMT and LT for
respectively IASI+GOME2 and IASI. In panel (e), DOFs for IASI+GOME2 are multiplied by
a factor 2 for visual clarity. **(g)** Potential vorticity at 300 (squares) and 500 hPa (triangles)
from ERAI reanalysis. **(h)** $NO_2$ concentrations observed as total columns by GOME-2 at 9h30
LT (squares) and OMI at 13h30 LT (triangles) and derived from CHASER analyses (stars) at
the LMT at 10h00 LT. In panels (c-h), curves with lighter and darker colours but the same
marker correspond to respectively the southern and northern pollution plumes Curves show
mean and standard deviations (+/- vertical bars) over the areas depicted by rectangles in Figs.
5 and 7-11 for each of the days of the pollution outbreak.