# Peer review of "Juan Cuesta1, Yugo Kanaya2, Masayuki Takigawa2, Gaëlle Dufour1, Maxim Eremenko1, Gilles Foret1, Kazuyuki Miyazaki2 and Matthias Beekmann1"

_Atmospheric Chemistry and Physics, 2017_

## Referee Comment (RC1) · Anonymous Referee #1 · 23 Nov 2017

General Comment

The authors examined a temporal evolution of ozone in a transboundary pollution event occurred in early May 2009 over East Asian countries by using multiple satellite observations and chemical transport models. The use of multispectral satellite data of IASI and GOME2 provides LMT ozone concentration which cannot be obtained by single-band retrievals. They clearly showed how well the IASI+GOME2 approach retrieve the ozone concentration in MLT and applied it to describe an outbreak of transboundary ozone pollution event in East Asia. The large-scale observation of ozone near the sur-

face (LMT) from the satellite with this approach is apparently a powerful tool for the air quality researches over the globe. This paper is well within the scope of the journal, however, I noticed several issues in this paper which cannot be passed over to be published. I suggested that the authors should consider the following comments: one major and several specific comments.

Major Comment:

My biggest concern is the arbitrariness in the use of model results. The authors used the simulation with two different chemistry transport models to explain the daily evolution of ozone pollution across East Asia. I suppose that both of the two models can simulate the same chemical quantities such as the three-dimensional concentration of $O_3$, $CO$, $NO_2$. However, the authors did not fully utilize the results from both models, but they only used the result from one of the models for one quantity in most cases. I don't think it is fair to arbitrary pick up only the better and propitious result from one of the models for their interpretation. I strongly suggest the authors to evenly use the results from both models for each chemical quantity. To put my point differently, the author should clearly state the different roles of the different models that they supposed at the very first part of the paper, so that the reader may not feel arbitrary use of the models.

Specific Comments:

- Abstract: The definition of lowermost tropospheric ozone should be provided.

- P3 L18-19: A brief explanation why single-band retrieval cannot provide the information in PBL is better here, not mandatory though.

-P6 L18-19: I could not find IASI+GOME2 data provided in both URL.

- P6 L28: Should more clearly describe the criteria of special coincidence. Is one degree lat/lon criteria between the location of sonde station and the center point of the satellite visual field?

- Table 1: This table is not referred in the manuscript. If it is not necessary for the paper, it's better to remove it.

- Figure 1: Is the symbol "$1\sigma1$" in legend widely used? I think at least a brief explanation is necessary in the caption.

- P7 L19: Why the sonde stations in Asia outside Japan, such as Hong Kong or Hanoi, were ignored? They have data for the year 2009 and 2010.

- P7 L28: EANET and GAW are different NW, so you should refer to them separately here. Also you should provide the source URL for GAW database.

- P8 L10: Is "vertical difference" more appropriate to this quantity than "vertical gradient" here. If you prefer to use gradient, it is natural to me that the unit is ppb/km.

- P8 L18-19: The author can discuss about the following paper here, since the paper also tried to retrieve the lower tropospheric ozone from OMI date.

Hayashida S., Liu X., Ono A., Yang K., Chance K., 2015. Atmospheric Chemistry and Physics 15, 9865-9881.

- Figure 3: Fig3a and b shows the LMT O3 in unit of ppb. Is this an average concentration of O3 in the lowermost 3 km altitude? If so, you should clearly state it somewhere in the manuscript (and the caption of this figure).

- P12 L6-10: The version of each EI should be described.

- P12 L12: Is this an appropriate reference for CHASER model?

- P12 L23-25: Is this reduction of bias for column density? How about the reduction of bias for the surface concentration by assimilation?

- P13 L2-4: Did you do assimilation for the sensitivity analysis too? If so, can we assume the influence of assimilation process is the same for both full and sensitivity simulation? If the impact of assimilation is different in two simulation, the difference of

ozone concentration between full and sensitivity simulation cannot be regarded as a pure stratospheric contribution.

- P13 L8: Is this reanalysis (ERA-INT) used to drive CHASER?

- P14 L16: You should describe clearly how to set the magenta rectangle in the figures. Are there any objective criteria to draw the four sides of the rectangle? There is no description of the rectangle in the caption of Figure 4 and 6.

- P16 L9-10: Where is "this location" here?

- P16 L16-17: I cannot agree with here, for me, Fig 6b is not so good agreement with Fig 5a.

- P16 L23-25: Which figure does this sentence mention to?

- P17 L26: Should clarify how to initiate the Hysplit calculation for two pollution plumes.

- P17 L27: Typo? (Fig. 9a)

- P18 L17: I cannot see the enhanced NO2 at the area pointed out here. Is this sentence correct?

- Figure 10e: The altitude of PV contour (300 hPa) should be described in the caption.

- P19 L21 & L28: Typo? 11e and 12e ?

- P20 L1: 11f and 12f?

- P21 L5: Fig 13 should be Fig 13a.

- P22 L4: NO titration

- P22 L6: Only horizontal dilution is important?

- P22 L12: I don't think the absolute values of the ratio dO3/dCO are consistent with each other from satellites observations and models. The models show apparent lower ratio than the observation. Please clarify what aspect is consistent each other.

- P22 L14: Typo? "those from models"?

- P22 L21: It is not easy to understand the meaning and implication of "degrees of freedom" for the readers outside the satellite data analysis. It's better to briefly explain them here.

- P22 L 25-26: Which figure does this sentence mention to?

- P22 L29: What is "3-9" here?

- P23 L1-2: I don't think the stratospheric contribution remains constant in anyway. It fluctuates a lot during the period.

- P23 L7-10: It is not easy to see consistency in the ozone partial column between satellite observation and model (CHAER) simulation. It's better to use more words to describe which aspect do you think is consistent to each other.

- P23 L16-17: How does the concentration change in $NO_2$ on 7 May simulated by WRF-Chem? Is it similar to what simulated in CHASER?

- P23 L23: Typo. red curve -> blue curve

- P24 L2-5: The authors referred to relatively high $NO_2$ concentration in CHASER as a cause of greater growth of $dO_3/dCO$ ration in the northern plume than the southern plume from 7 to 9 May. However, the ration in CHASER did not show such a growth both in the northern and southern plumes (Figure 13c). So it is not appropriate to refer to the change in $NO_2$ in CHASER as a cause of the observed change in the ratio.

- P24 L17-19: Can you estimate the impact of this effect on the $dO_3/dCO$ ratio quantitatively? Excluding under- or over-estimation of $dO_3/dCO$ ratio due to the change in the sensitivity of satellite retrievals is quite important to make this approach useable for $O_3$ production estimation during air mass transport

---

## Referee Comment (RC2) · Anonymous Referee #2 · 3 Mar 2018

This paper elaborates a comprehensive study of transboundary ozone pollution across East Asia via employing [1] the multiple-spectral IASI/GOME2 ozone profile product that provides the quantitative estimates of ozone concentration in the LMT; and [2] the combined modeling tools consisting of CHASER (global scale) and WRF-CHEM (regional scale) models. This study provides multi-species, multi-scale picture of pollutions across East Asia, helping in distinguishing between local and non-local drivers of pollution in LMT. The subject of the paper is appropriate to ACP. Below are a few comments concerning clarifications/extensions for consideration in the final publication

in ACP.

This work uses the IASI carbon monoxide (CO) profile data to estimate the CO concentration in lower troposphere (LT), then use IASI LT CO and joint IASI+GOME2 LMT O3 as daily pictures for facilitate the study of daily evolution of pollution across East Asia. The authors should describe how well the IASI LT CO data could represent the CO variability in the LMT.

Drs. Miyazaki and Sekiya have developed a high-resolution CHASER simulation tool (version 4.0) with a finest spatial resolution of 0.56 degrees (Sekiya et al., 2017) – significantly higher that of CHASER and WRF-Chem models used in this study. The performance of CHASER v4 has been validated using reference data sets from satellite missions and aircraft flight campaigns. The authors should include this reference in this paper and provides some discussions. Sekiya T., Miyazaki K., Ogochi K., Sudo K., and Takigawa M., Global high-resolution simulations of tropospheric nitrogen dioxide using CHASER V4.0, Geosci. Model Dev. Discuss., https://doi.org/10.5194/gmd-2017-203, in review, 2017.

Page 4, Line 5-6: There is a multiple spectral retrieval algorithm developed for CO profile retrievals (Fu et al. 2016). They demonstrated the feasibility of combining the measurements from Sentinel-5 precursor (S5P) TROPOMI (near infrared) and Suomi-NPP (SNPP) CrIS (thermal Infrared) sensors to extend Terra MOPIIT both TIR alone and multiple spectral CO profile products capable of quantifying the first 2-3 km CO amounts, as well as improving spatial coverage and resolution in comparison to Terra-MOPITT. The authors could add some discussions nearby the end of first/beginning of second paragraphs of page 4, e.g, "The Sentinel-5 precursor (S5P) and Suomi NPP (SNPP) has successfully formed a new satellite constellation, leading to a unique opportunity to quantify the amounts of carbon monoxide in the LMT over global scale via combining the satellite measurements from SNPP CrIS (TIR) and S5P TROPOMI (NIR) instruments. Fu et al. (2016) presented the methodology and characteristics of joint CrIS/TROPOMI CO profile retrievals, demonstrating the feasibility of extending the

decadal record of Terra-MOPITT CO products (Worden et al., 2010 and 2013)." Fu D., Bowman K.W., Worden H., Natraj V., Yu S., Worden J.R., Veefkind P., Aben I., Landgraf J., Strow L., Han Y., High resolution tropospheric carbon monoxide profiles retrieved from CrIS and TROPOMI, Atmos. Meas. Tech., 9, 2567-579, 2016.

Worden H.M., Deeter M.N., Edwards D.P., Gille J.C., Drummond J. R., and Nédélec, P. P., Observations of near-surface carbon monoxide from space using MOPITT multi-spectral retrievals, J. Geophys. Res., 115, D18314, doi:10.1029/2010JD014242, 2010.

Worden H.M., Deeter M.N., Frankenberg C., George M., Nichitiu F., Worden J., Aben I., Bowman K. W., Clerbaux C., Co-heur P.F., de Laat A.T.J., Detweiler R., Drummond J. R., Edwards D.P., Gille J. C., Hurtmans D., Luo M., Martínez-Alonso S., Massie S., Pfister G., and Warner J.X., Decadal record of satellite carbon monoxide observations, Atmos. Chem. Phys., 13, 837–850, doi:10.5194/acp-13-837-2013, 2013.

Page 38, Line 2, Figure 2 caption: IASI+GOME -> IASI+GOME2

---

## Author Comment (AC1) · 2 May 2018

The supplement file acp-2017-972-supplement.pdf provides identical answers to the ones below, but in colored formatted text.

Dear referees, We would like to thank you very much for your remarks that have improved the clarity of the paper. In the Revised Manuscript, called RM hereafter, we have addressed in detail each of your comments by adding new explanations in the manuscript and some minor modifications in the figures. All the recommendations of

the reviewers have been followed and all clarifications were provided. Please, find below the detailed answers and how they are introduced in the manuscript.

Anonymous Referee #1

General Comment The authors examined a temporal evolution of ozone in a transboundary pollution event occurred in early May 2009 over East Asian countries by using multiple satellite observations and chemical transport models. The use of multispectral satellite data of IASI and GOME2 provides LMT ozone concentration, which cannot be obtained by single-band retrievals. They clearly showed how well the IASI+GOME2 approach retrieve the ozone concentration in LMT and applied it to describe an outbreak of transboundary ozone pollution event in East Asia. The large-scale observation of ozone near the surface (LMT) from the satellite with this approach is apparently a powerful tool for the air quality researches over the globe. This paper is well within the scope of the journal, however, I noticed several issues in this paper, which cannot be passed over to be published. I suggested that the authors should consider the following comments: one major and several specific comments.

Major Comment: My biggest concern is the arbitrariness in the use of model results. The authors used the simulation with two different chemistry transport models to explain the daily evolution of ozone pollution across East Asia. I suppose that both of the two models can simulate the same chemical quantities such as the three-dimensional concentration of O3, CO, NO2. However, the authors did not fully utilize the results from both models, but they only used the result from one of the models for one quantity in most cases. I don't think it is fair to arbitrary pick up only the better and propitious result from one of the models for their interpretation. I strongly suggest the authors to evenly use the results from both models for each chemical quantity. To put my point differently, the author should clearly state the different roles of the different models that they supposed at the very first part of the paper, so that the reader may not feel

arbitrary use of the models.

Clarified and completed.

This aspect is fully clarified in the RM and Figure 14 is completed for consistency. Three-dimensional fields of O3, CO and NO2 are available from both WRF-Chem and CHASER models. However, using both models in all cases would largely increase the number of figures, without adding much relevant information for the paper. Therefore, the criteria to choice between the models used in the figures of the paper are the following: i) WRF-Chem is used to describe the structure of the plumes of LMT ozone and carbon monoxide with finer spatial resolution (Figures 4c-f, 6a-c, 10d, 11d and 12d), ii) CHASER forecasts with and without stratospheric ozone allows a distinction between tropospheric ozone originating from the stratosphere and the troposphere (Figs. 6e-f, 10f, 11f, 12f, 14e-f), iii) both WRF-Chem and CHASER analyses are used for showing the temporal lagrangian evolution of O3, CO and NO2 for the polluted air masses (Figure 13b-c and 14a-d,j) and iv) in the comparison of IASI+GOME2 and in situ data (Figure 2), CHASER analyses indicate the corresponding vertical gradient of ozone between the surface and 2 km of altitude with presumably good absolute accuracy provided by assimilation of various observations. For consistency in Fig. 14, we have added the time series of O3, CO and NO2 concentrations (Figs. 14c, d and j respectively) from CHASER analyses.

This clarification is added in the RM as follows (lines 28-32, page 12, lines 1-4, page 13): "In the figures of the paper, we show one of the models or both of them according to the following criteria: i) WRF-Chem describes the structure of plumes of LMT O3 and CO with finer spatial resolution (section 3), ii) CHASER forecasts with and without stratospheric ozone distinguish tropospheric ozone formed at the troposphere from that originating from the stratosphere (sections 3 and 4), iii) both WRF-Chem and CHASER are used for showing the temporal Lagrangian evolution of O3, CO and NO2 for polluted air masses (section 4) and iv) CHASER analyses indicate the vertical gradients of ozone between the surface and 2 km of altitude with presumably good absolute

accuracy provided by assimilation of various observations (section 2)."

Specific Comments: - Abstract: The definition of lowermost tropospheric ozone should be provided.

Done.

The definition is provided as (line 16 of page 1): " ... lowermost tropospheric ozone (located below 3 km of altitude)"

- P3 L18-19: A brief explanation why single-band retrieval cannot provide the information in PBL is better here, not mandatory though.

Clarified.

The following brief clarification is provided in the RM (lines 19-23 of page 3): "Standard single-band ozone retrievals cannot provide quantitative information at the planetary boundary layer (PBL), but at lowest at the lower troposphere (LT, i.e. below 6 km of altitude). Sensitivity to ozone for these retrievals essentially peaks at the free troposphere above the PBL, according to the available information on near-surface ozone."

-P6 L18-19: I could not find IASI+GOME2 data provided in both URL.

Corrected. The URL has been updated.

You can now find a description of the IASI+GOME2 data and the way to obtain it at http://cds-espri.ipsl.fr The general portal of the data centre https://www.aeris-data.fr redirects the user to http://cds-espri.ipsl.fr . It is important to mention both URLs. Data availability is better described in the RM as (line 32, page 6 and lines 1-2, page 7): "global scale IASI+GOME2 retrievals are routinely produced by the French data centre AERIS and they are publicly available (see https://www.aeris-data.fr and http://cds-espri.ipsl.fr)."

- P6 L28: Should more clearly describe the criteria of special coincidence. Is one degree lat/lon criteria between the location of sonde station and the center point of the

satellite visual field?

Clarified.

Yes, it is one-degree latitude/longitude between the location of launching station of the sonde and the centre point of the satellite pixel.

This is clarified in the RM as (lines 11-13 page 7): "Coincidence criteria are spatial co-localization of one-degree latitude/longitude between the locations of the launching stations of the sondes and the centre points of satellite pixels".

- Table 1: This table is not referred in the manuscript. If it is not necessary for the paper, it's better to remove it.

Corrected.

Table 1 is useful for the paper as it allows the reader to quickly and easily read the results of the comparison between ozonesondes and the satellite retrieval.

The reference to Table 1 has been added in the RM as (lines 4-5 page 7): "An assessment of the quality of IASI+GOME2 for retrieving LMT ozone is presented in Figure 1 and summarized in Table 1".

- Figure 1: Is the symbol "$1\sigma1$" in legend widely used? I think at least a brief explanation is necessary in the caption.

Clarified.

For clarity in the RM, the symbol has been changed to $\sigma$sonde/$\sigma$sat and the captions of Figure 1 and 2 explain it as: "The symbol $\sigma$sonde/$\sigma$sat is the ratio between the standard deviations of the sonde data and the satellite retrievals."

- P7 L19: Why the sonde stations in Asia outside Japan, such as Hong Kong or Hanoi, were ignored? They have data for the year 2009 and 2010.

Clarified.

Sonde data from Hong Kong and Hanoi stations is included in the Worldwide comparison (Fig. 1a and Table 1). The region of Asia analysed in the paper is 25-45°N 110-150°E. Hong Kong and Hanoi are located south of 25°N and therefore they were not included in the East Asian analysis.

This aspect in clarified in the RM as (line 32, page 7 and line 1 page 8): "As this paper focuses on East Asia (particularly at 25-45°N 110-150°E), we also present the comparison for all sondes available over this region ... "

- P7 L28: EANET and GAW are different NW, so you should refer to them separately here. Also you should provide the source URL for GAW database.

Corrected.

Both networks are referred in RM separately as follows (lines 11-14 page 8): "from 9 stations of the EANET (Acid Deposition Monitoring Network in East Asia, http://www.eanet.asia) network over East Asia, one station from the GAW (Global Atmosphere Watch, http://www.wmo.int) network"

- P8 L10: Is "vertical difference" more appropriate to this quantity than "vertical gradient" here. If you prefer to use gradient, it is natural to me that the unit is ppb/km.

Corrected.

We use in the RM the unit ppb/km for the vertical gradient, as follows (lines 26-27 page 8): "vertical gradients $\Delta$O3surf.-2km between the surface and 2 km of altitude lower than $\pm$ 10 ppb/km"

- P8 L18-19: The author can discuss about the following paper here, since the paper also tried to retrieve the lower tropospheric ozone from OMI data. Hayashida S., Liu X., Ono A., Yang K., Chance K., 2015. Atmospheric Chemistry and Physics 15, 9865-9881.

Done.

We have mentioned the lower tropospheric ozone retrieval from OMI in this paragraph of the RM as follows (lines 22-25 page 9): "This IASI retrieval is often used to analyse ozone enhancements at the lower troposphere over Europe (e.g. Eremenko et al., 2008) and East Asia (e.g. Dufour et al., 2015), as also done with retrievals from OMI measurements also over East Asia (Hayashida et al., 2015)."

- Figure 3: Fig3a and b shows the LMT O3 in unit of ppb. Is this an average concentration of O3 in the lowermost 3 km altitude? If so, you should clearly state it somewhere in the manuscript (and the caption of this figure).

Clarified.

In the text of the RM and the caption of Fig. 2, we clearly state is as (line 32 page 8 and lines 1-3 and page 8): "Ozone concentrations at the LMT are provided as volume mixing ratios in ppb (parts per billion), calculated as the ratio of LMT partial columns (in molecules per cm2) of ozone and air (in Figures 2-14 and also used for CO and other partial columns)."

- P12 L6-10: The version of each EI should be described.

Done.

In the RM, we have added the version of each Emission inventory (lines 10-13 page 13): "Surface emissions over China and North and South Korea are taken from REAS (Regional Emission Inventory in Asia; Ohara et al. 2007) version 1.11, and over Russia from EDGAR (Emission Database for Global Atmospheric Research; Olivier et al. 1996) version 3.2."

- P12 L12: Is this an appropriate reference for CHASER model?

Corrected.

For the CHASER model, the RM use the following references (lines 16-17 page 12): "Sudo et al., 2002; Sudo and Akimoto, 2007; Sekiya and Sudo, 2014"

Sudo, K., Takahashi, M., Kurokawa, J., and Akimoto, H.: CHASER: A global chemical model of the tropo- sphere 1. Model description, J. Geophys. Res., 107, 4339, https://doi.org/10.1029/2001JD001113, 2002.

Sudo, K. and Akimoto, H.: Global source attribution of tropospheric ozone: Long-range transport from various source regions, J. Geophys. Res., 112, D12302, https://doi.org/10.1029/2006JD007992, 2007.

Sekiya, T. and Sudo, K.: Roles of transport and chemistry processes in global ozone change on interannual and multidecadal time scales, J. Geophys. Res., 119, 4903–4921, https://doi.org/10.1002/2013JD020838, 2014.

- P12 L23-25: Is this reduction of bias for column density? How about the reduction of bias for the surface concentration by assimilation?

Clarified.

The reductions in biases by data assimilation are enounced in column for NO2 and concentrations for CO in the lower troposphere and O3 in the middle and upper troposphere.

This aspect is fully clarified in the RM as (lines 25-31 page 13): "It reduces biases for tropospheric NO2 columns by 40–85 %, for lower tropospheric CO concentrations in the Northern Hemisphere by 40–90 % and for O3 in the middle and upper troposphere by 30–40%. Data assimilation also mostly removed the model's negative bias in surface CO concentrations in the northern hemisphere. The error reduction for O3 was generally smaller in the lower troposphere than in the middle and upper troposphere because of the reduced sensitivity of the assimilated TES retrievals to lower-tropospheric ozone."

- P13 L2-4: Did you do assimilation for the sensitivity analysis too? If so, can we assume the influence of assimilation process is the same for both full and sensitivity simulation? If the impact of assimilation is different in two simulations, the difference

of ozone concentration between full and sensitivity simulation cannot be regarded as a pure stratospheric contribution.

Clarified.

Assimilation is not performed in the simulations used in the sensitivity analysis. The difference is calculated between two simulations run in forecast mode. The difference between the 2 simulations is therefore consistent for estimating tropospheric and stratospheric contributions of ozone.

The RM clarifies this issue as (lines 21-22 page 14): "For consistency, no data assimilation is performed in either of the two simulations of this sensitivity analysis."

- P13 L8: Is this reanalysis (ERA-INT) used to drive CHASER?

Clarified.

CHASER is driven by AGCM meteorological fields nudged toward NCEP-DOE/AMIP-II reanalyses at every time step of the AGCM to reproduce past meteorological fields.

This aspect is indicated in the RM as (lines 9-12 page 14): "The AGCM fields are nudged toward the National Centers for Environmental Prediction/Department of Energy Atmospheric Model Intercomparison Project II (NCEP-DOE/AMIP-II) reanalysis (Kanamitsu et al., 2002) at every time step of the AGCM to reproduce past meteorological fields."

- P14 L16: You should describe clearly how to set the magenta rectangle in the figures. Are there any objective criteria to draw the four sides of the rectangle? There is no description of the rectangle in the caption of Figure 4 and 6.

Clarified.

Magenta and red rectangles in Figures 4-12 are rectangular zones containing all valid satellite pixels (gridded $1° \times 1°$) used to describe the daily evolution of polluted air masses originating from the North China Plain and travelling to the Pacific. These

satellite pixels are those co-located with at least 5 % of the polluted air parcels trajectories simulated by HYSPLIT dispersion model.

In the RM, this is clarified as follows (lines 16-19 page 16): "In Figures 4-12, satellite pixels used to describe the evolution of these polluted air masses are depicted by magenta and red rectangles. These boxes contain valid satellite pixels co-located with at least 5 % of the polluted air parcels trajectories simulated by Hysplit."

- P16 L9-10: Where is "this location" here?

Clarified.

This statement on Fig. 6b refers to the region 40-45°N 122-128°E (previously commented for Fig. 5b).

The RM indicates is as (line 31 page 17): "At this location (40-45°N 122-128°E)"

- P16 L16-17: I cannot agree with here, for me, Fig 6b is not so good agreement with Fig 5a.

Corrected.

Indeed, differences in LMT ozone simulated by WRF-Chem and retrieved by IASI+GOME2 are seen north of 38°N (110-120°E) with higher values for satellite retrievals. However, both WRF-Chem and IASI+GOME2 show relatively high LMT ozone concentrations over the NCP.

This is corrected in the RM as (lines 6-8 page 18): "As previously mentioned, IASI+GOME2 also retrieves high LMT ozone concentrations over the NCP (around 35°N 115°E, Fig. 5a), but also north of it (differing from simulations at the LMT)."

- P16 L23-25: Which figure does this sentence mention to?

Clarified and Corrected.

This sentence refers to the general method used for all figures (Figure 4-12) where the

polluted air masses are tracked. In this paragraph, it is indeed misleading. As it is also redundant with section 2.4, the referred sentence is erased from the RM.

- P17 L26: Should clarify how to initiate the Hysplit calculation for two pollution plumes.

Clarified.

In our analysis, we initialize Hysplit at two starting locations separated by a relatively small distance, which suggests two different trajectories for the polluted air masses. The trajectory of the southern plume (magenta in Fig. 9) is obtained with the mean location of the air parcels of the previous day and that of the northern plume (red in Fig. 9) by initializing Hysplit 2 degrees northeast from that position. The existence of two different LMT ozone plumes is clearly evidenced by IASI+GOME2 observations in the following days (Fig. 12) and is also confirmed by trajectories in backward mode initiated at the location of these plumes.

In the RM, we clarify this aspect as (lines 21-27 page 19): "The trajectories of the southern and northern pollution plumes (respectively magenta and red in Fig. 9) are obtained by initializing Hysplit respectively at the mean arrival location of the trajectories from the previous day and 2 degrees northeast from that position. The common geographical origin of the two plumes (before 6 May) is confirmed by Hysplit trajectories in backward mode initiated at the location of the two major ozone plumes clearly observed by IASI+GOME2 (e.g. south and north of Japan two days after in Fig. 12a)."

- P17 L27: Typo? (Fig. 9a)

Corrected.

Both Fig.9a and 9c show the two depicted pollution plumes, respectively with boxes and the actual location of the Hysplit air parcels. In the RM, we indicate it as (line 17 page 19): "Fig. 9a,c".

- P18 L17: I cannot see the enhanced NO2 at the area pointed out here. Is this sentence correct?

Clarified.

The enhancement of NO2 with respect to the background at the region 43-45°N 126-132°E is only moderate (up to 6 1015 mol/cm2). Therefore, the sentence is changed in the RM as (lines 14-15 page 20): "as suggested by moderately enhanced NO2 concentrations at 43-45°N 126-132°E up to 6 1015 mol/cm2 in Fig. 10c".

- Figure 10e: The altitude of PV contour (300 hPa) should be described in the caption.
- P19 L21 & L28: Typo? 11e and 12e ?

Corrected.

In the RM, we have added "at 300 hPa" in the caption of Figure 10e and corrected the typos on the figures. For CHASER, we indicate Fig.11f and Fig.12f and for PV Fig.11e and Fig.12e.

- P20 L1: 11f and 12f?

Corrected.

The RM indicates Fig. 11f and 12f.

- P21 L5: Fig 13 should be Fig 13a.

Corrected.

The RM indicates Fig. 13a (line 5 page 23).

- P22 L4: NO titration

Corrected.

The RM indicates (line 4 page 24) "nitrogen monoxide (NO) titration".

- P22 L6: Only horizontal dilution is important?

Clarified.

Since the reduction in CO concentrations are remarked during only 3 days and the typical lifetime of CO is in the order of a few months, the only remaining factor for reducing CO at the LMT is atmospheric dilution (horizontal and/or vertical). Since it is not only horizontal dilution, the statement in the RM has been changed to (lines 8-9 page 24): "... linked to atmospheric dilution (horizontal and/or vertical). Sinks of CO are not expected to be significant during a period of 3 days."

- P22 L12: I don't think the absolute values of the ratio dO3/dCO are consistent with each other from satellites observations and models. The models show apparent lower ratio than the observation. Please clarify what aspect is consistent each other.

Agreed and corrected.

We agree with your remark that in absolute values, the ratio $\Delta O3/\Delta CO$ derived from the satellite measurements is higher than that from models, particularly at the beginning of the period (before 6 May). This is modified in the RM (lines 13-17 page 24): "In absolute values, the ratio $\Delta O3/\Delta CO$ derived from the satellite measurements is higher than that from models. At the beginning of the event (3-5 May), satellite estimates of the ratio are 0.1 to 0.15 higher than those from satellite. After 6 May, satellite and WRF-Chem ratios are closer (with differences between 0.05 and 0.1)."

- P22 L14: Typo? "those from models"?

Corrected.

Indeed, it is "those from models" as corrected in the RM (line 16 page 24).

- P22 L21: It is not easy to understand the meaning and implication of "degrees of freedom" for the readers outside the satellite data analysis. It's better to briefly explain them here.

Clarified.

The following description of the "degrees of freedom" of the satellite retrievals is provided in the RM (line 10 page 3): "degrees of freedom (i.e. the number of independent pieces of information of the retrieved profile)" and (lines 25-28 page 24): "This is described in terms of the degrees of freedom (i.e. the number of independent pieces of information) and the altitude of maximum sensitivity of the retrieved atmospheric columns, which respectively quantify the amount of information provided by the satellite retrieval and the altitude it comes from."

- P22 L 25-26: Which figure does this sentence mention to? - P22 L29: What is "3-9" here?

Clarified.

This sentence comes from Figure 14h of the RM. The word "May" is missing after "3-9". These corrections are included in the RM (line 5 page 25).

- P23 L1-2: I don't think the stratospheric contribution remains constant in anyway. It fluctuates a lot during the period.

Agreed and corrected.

We agree that the stratospheric contribution at the upper troposphere fluctuates significantly. This mistake was corrected in the RM as (lines 11-13 page 25): "On the other hand, the ozone contribution of stratospheric downward transport at the upper troposphere (from 6 to 12 km asl) fluctuates significantly during the whole event (Fig. 14f)."

- P23 L7-10: It is not easy to see consistency in the ozone partial column between satellite observation and model (CHASER) simulation. It's better to use more words to describe which aspect do you think is consistent to each other.

Clarified.

A fair consistency between IASI+GOME2 and CHASER is remarked in the average concentrations of ozone at the LMT and the upper troposphere. We clarify it in the RM

as (lines 15-18 page 25): "we remark that similar concentrations of ozone at the LMT and the upper troposphere are retrieved by IASI+GOME2 and simulated by CHASER (adding contributions from the Troposphere and Stratosphere in) averaged over the whole event (differences of 13 ppb at most)."

- P23 L16-17: How does the concentration change in NO2 on 7 May simulated by WRF-Chem? Is it similar to what simulated in CHASER?

Clarified.

Yes, as shown in the new figure 14j of the RM, NO2 concentrations simulated by both WRF-Chem and CHASER on 7 May show very similar relative temporal evolutions, with a clear enhancement with respect to the previous day. The similarity in the evolution of NO2 simulated by both model is kept over whole period (3-9 May) in relative terms, but LMT NO2 concentrations in absolute values are a factor 3 higher for WRF-Chem than for CHASER.

In the RM, this is written as (lines 9-11 page 26): "Both WRF-Chem and CHASER simulations suggest a relatively higher availability of NO2 at the LMT for the northern pollution filament" and (lines 4-5 page 26) "The latter might be linked to low availability of NO2 at the LMT in CHASER simulations (a factor 3 lower than for WRF-Chem, Fig. 14j)"

- P23 L23: Typo. red curve -> blue curve

Agreed and corrected.

- P24 L2-5: The authors referred to relatively high NO2 concentration in CHASER as a cause of greater growth of dO3/dCO ratio in the northern plume than the southern plume from 7 to 9 May. However, the ratio in CHASER did not show such a growth both in the northern and southern plumes (Figure 13c). So it is not appropriate to refer to the change in NO2 in CHASER as a cause of the observed change in the ratio.

Agreed and clarified.

We agree that only CHASER simulations are not sufficient evidence to analyse the observed growth of dO3/dCO. However, WRF-Chem shows the same relative increase as CHASER in NO2 concentrations on 7-9 May, and higher by a factor 3. Therefore, we mention the enhancement of LMT NO2 for both WRF-Chem and CHASER simulations (3 times higher for WRF-Chem) that suggest a higher availability of NO2 to explain the observed change in the ratio. In the RM, this is written as (lines 9-13 page 26): "Both WRF-Chem and CHASER simulations suggest a relatively higher availability of NO2 at the LMT (although 3 times higher for WRF-Chem) for the northern pollution filament (dotted curves with respectively blue ovals and green stars in Fig. 14j) as for the southern plume (light blue and light green in Fig. 14j)."

- P24 L17-19: Can you estimate the impact of this effect on the dO3/dCO ratio quantitatively? Excluding under- or over-estimation of dO3/dCO ratio due to the change in the sensitivity of satellite retrievals is quite important to make this approach useable for O3 production estimation during air mass transport.

Done and clarified.

We have done a sensitivity analysis of the errors in dO3/dCO with respect to changes in sensitivity of the satellite retrievals. For this, we have used model outputs of typical vertical profiles of O3 and CO for a pollution plume and smoothed them with typical averaging kernels of the satellite retrievals. We have taken into account that heights of maximum sensitivity usually change concomitantly for both O3 and CO retrievals, as in both cases it depends on thermal contrast between the surface and the air. The results show that under and overestimations of dO3/dCO remain below +/- 11 % for changes of 1 and 3 km in the heights of maximum sensitivity for respectively O3 and CO. This uncertainty is significantly lower than changes observed for dO3/dCO from satellite retrievals (up to 80% during the whole event). Therefore, the proposed approach is valid for estimating O3 production during air mass transport.

These results and clarifications are provided in the RM as (lines 27-32 page 26 and

lines 1-5 page 27): "According to sensitivity analyses, these uncertainties induce under or overestimations for $\Delta O3/\Delta CO$ that remain below $\pm$ 11 % for changes of 1 and 3 km in the heights of maximum sensitivity for respectively O3 and CO retrievals. These estimations are obtained using typical vertical profiles of O3 and CO for a pollution plume (from WRF-Chem) smoothed with averaging kernels of the satellite retrievals and taking into account the concomitant change in the heights of maximum sensitivity for O3 and CO retrievals, as in both cases they depend on the difference between surface and air temperatures. These uncertainties are significantly lower than changes observed for $\Delta O3/\Delta CO$ from satellite retrievals (up to 84% during the whole event). Therefore, conclusions drawn on the occurrence and quantification of photochemical ozone production in this period are not significantly affected by changes in satellite retrievals sensitivities."

Please also note the supplement to this comment: https://www.atmos-chem-phys-discuss.net/acp-2017-972/acp-2017-972-AC1-supplement.pdf

---

## Author Comment (AC2) · 2 May 2018

The supplement file acp-2017-972-supplement.pdf of this comment provides identical answers to the ones below, but in colored formatted text.

Dear referees, We would like to thank you very much for your remarks that have improved the clarity of the paper. In the Revised Manuscript, called RM hereafter, we have addressed in detail each of your comments by adding new explanations in the manuscript and some minor modifications in the figures. All the recommendations of

the reviewers have been followed and all clarifications were provided. Please, find below the detailed answers and how they are introduced in the manuscript.

Anonymous Referee #2

This paper elaborates a comprehensive study of transboundary ozone pollution across East Asia via employing [1] the multiple-spectral IASI/GOME2 ozone profile product that provides the quantitative estimates of ozone concentration in the LMT; and [2] the combined modeling tools consisting of CHASER (global scale) and WRF-CHEM (regional scale) models. This study provides multi-species, multi-scale picture of pollutions across East Asia, helping in distinguishing between local and non-local drivers of pollution in LMT. The subject of the paper is appropriate to ACP. Below are a few comments concerning clarifications/extensions for consideration in the final publication. This work uses the IASI carbon monoxide (CO) profile data to estimate the CO con- centration in lower troposphere (LT), then use IASI LT CO and joint IASI+GOME2 LMT O3 as daily pictures for facilitate the study of daily evolution of pollution across East Asia.

1) The authors should describe how well the IASI LT CO data could represent the CO variability in the LMT.

Clarified.

The approach developed by ULB/LATMOS retrieves lower tropospheric CO (below 6 km of altitude) from IASI with 0.83 degrees of freedom and a height of maximum sensitivity around 4.7 km of altitude asl, in average over the region and period studied in the paper. This CO product is sensitive at the LMT (up to 3 km of altitude asl) with 0.51 degrees of freedom in average. In our paper, we use LT CO retrievals as they are already validated against MOZAIC in situ measurements (De Wachter et al., 2012). A dedicated validation of other CO partial columns against independent measurements

is beyond the scope of the present paper, as it is mainly focused on ozone pollution.

In the RM, this is written as (lines 2-5 page 11): "LT partial columns are retrieved with 0.83 degrees of freedom (DOF, i.e. number of independent pieces of information in the retrieved profile) in average over the region and period studied in the paper. This product provides significant information on CO variability below 3 km asl, as DOF at the LMT are 0.51 on average."

2) Drs. Miyazaki and Sekiya have developed a high-resolution CHASER simulation tool (version 4.0) with a finest spatial resolution of 0.56 degrees (Sekiya et al., 2017) – significantly higher that of CHASER and WRF-Chem models used in this study. The performance of CHASER v4 has been validated using reference data sets from satellite missions and aircraft flight campaigns. The authors should include this reference in this paper and provides some discussions. Sekiya T., Miyazaki K., Ogochi K., Sudo K., and Takigawa M., Global high-resolution simulations of tropospheric nitrogen dioxide using CHASER V4.0, Geosci. Model Dev. Discuss., https://doi.org/10.5194/gmd-2017-203, in review, 2017.

Done.

Indeed, Sekiya et al. (2018) developed a high-resolution version of the CHASER model at 0.56° resolution and demonstrated the improved model performance over areas with strong local sources by increased the horizontal model resolution from 2.8° to 0.56°. Nevertheless, the 2.8° resolution model is capable to simulate synoptic ozone patterns.

The RM provides this information as follows (lines 2-6 page 14): "Sekiya et al. (2018) have recently developed a high-resolution version of CHASER with 0.56° horizontal resolution and demonstrated improved performances over areas with strong local sources with respect to the 2.8° resolution version. Nevertheless, the CHASER model with 2.8° resolution is capable of properly simulating synoptic ozone patterns."

3) Page 4, Line 5-6: There is a multiple spectral retrieval algorithm developed for CO

profile retrievals (Fu et al. 2016). They demonstrated the feasibility of combining the measurements from Sentinel-5 precursor (S5P) TROPOMI (near infrared) and Suomi-NPP (SNPP) CrIS (thermal Infrared) sensors to extend Terra MOPIIT both TIR alone and multiple spectral CO profile products capable of quantifying the first 2-3 km CO amounts, as well as improving spatial coverage and resolution in comparison to Terra-MOPITT. The authors could add some discussions nearby the end of first/beginning of second paragraphs of page 4, e.g, "The Sentinel-5 precursor (S5P) and Suomi NPP (SNPP) has successfully formed a new satellite constellation, leading to a unique opportunity to quantify the amounts of carbon monoxide in the LMT over global scale via combining the satellite measurements from SNPP CrIS (TIR) and S5P TROPOMI (NIR) instruments. Fu et al. (2016) presented the methodology and characteristics of joint CrIS/TROPOMI CO profile retrievals, demonstrating the feasibility of extending the decadal record of Terra-MOPITT CO products (Worden et al., 2010 and 2013)."

Fu D., Bowman K.W., Worden H., Natraj V., Yu S., Worden J.R., Veefkind P., Aben I., Landgraf J., Strow L., Han Y., High resolution tropospheric carbon monoxide profiles retrieved from CrIS and TROPOMI, Atmos. Meas. Tech., 9, 2567-579, 2016.

Worden H.M., Deeter M.N., Edwards D.P., Gille J.C., Drummond J. R., and NeÌĄdeÌĄlec, P. P., Observations of near-surface carbon monoxide from space using MOPITT multi-spectral retrievals, J. Geophys. Res., 115, D18314, doi:10.1029/2010JD014242, 2010.

Worden H.M., Deeter M.N., Frankenberg C., George M., Nichitiu F., Worden J., Aben I., Bowman K. W., Clerbaux C., Co-heur P.F., de Laat A.T.J., Detweiler R., Drummond J. R., Edwards D.P., Gille J. C., Hurtmans D., Luo M., MartiÌĄnez-Alonso S., Massie S., Pfister G., and Warner J.X., Decadal record of satellite carbon monoxide observations, Atmos. Chem. Phys., 13, 837–850, doi:10.5194/acp-13-837-2013, 2013.

Done.

The 3 references have been added as well as the following paragraph in the RM (lines

9-20 page 4): "Multispectral synergisms are also implemented to retrieve other atmospheric species with enhanced near-surface sensitivity, as carbon monoxide (CO). This is done with measurements in the thermal and near infrared from the Measurements Of Pollution In The Troposphere (MOPITT) instrument onboard the Earth Observing System (EOS) Terra satellite (Worden et al., 2010). Recently, Sentinel-5 precursor (S5P) and Suomi National Polar orbiting Partnership (SNPP) have successfully formed a satellite constellation, leading to a new opportunity to quantify the amounts of CO at the LMT over global scale by combining the satellite measurements in the thermal and near IR respectively from the instruments SNPP Cross-track Infrared Sounder (CrIS) and S5P TROPOspheric Monitoring Instrument (TROPOMI). Fu et al. (2016) presented the methodology and characteristics of joint CrIS/TROPOMI CO profile retrievals, demonstrating the feasibility for extending the decadal record of MOPITT CO products (Worden et al., 2013)."

4) Page 38, Line 2, Figure 2 caption: IASI+GOME -> IASI+GOME2

Corrected.

Please also note the supplement to this comment:
https://www.atmos-chem-phys-discuss.net/acp-2017-972/acp-2017-972-AC2-supplement.pdf